



# Warmer winters causes an increase of chlorophyll-a concentration in deeper layers: the opposite role of convection and self-shading on the example of the Black Sea

Elena A. Kubryakova, Arseny A. Kubryakov

Marine Hydrophysical Institute, Russian Academy of Sciences, Sevastopol, Russian Federation

*Correspondence to*: Elena A. Kubryakova (elena_kubryakova@mail.ru)

**Abstract.** Winter vertical entrainment of deep waters determines not only the amount of nutrients in the upper layers, but also the light conditions in it, through the self-shading mechanism. In this paper, we use Bio-Argo data to demonstrate significant differences in the vertical distribution of chlorophyll-a concentration (Chl) in the Black Sea between a year with cold winter

(2017) and a year with warm winter (2016). Stronger vertical entrainment of nutrient-rich waters from deeper isopycnal layers in cold 2017 caused an increase of Chl in winter up to 0.6-0.7 mg/m$^3$ compared to a warm winter of 2016, when Chl was only 0.4-0.5 mg/m$^3$. Further, during almost the whole year from February to October Chl in the upper 0-40 m layer of cold 2017 year was on 0.1-0.2 mg/m$^3$ higher than in 2016. This rise of Chl in 2017 led to an increase in light attenuation due to the self-shading effect. In contrast, in warm 2016 with a lower amount of nutrients light attenuation decreased and the irradiance

reached deeper isopycnals layers with a higher amount of nutrients. As a result, in warm 2016 the subsurface chlorophyll maximum deepens and the values of Chl in 40-60 m layers were significantly higher than in 2017. The maximum positive difference in this layer (0.5 mg/m$^3$) was observed during a summer seasonal peak of irradiance due to the largest increase of light attenuation in the summer of 2017. As a result, the column-averaged yearly values of Chl in warm 2016 and cold 2017 were comparable. However, in the year with intense winter mixing upper layers are more productive, while in the year with

low winter vertical mixing, subsurface chlorophyll maximum widens and reaches deeper layers. These results show that the observed long-term warming may lead to the continuous deepening of the subsurface chlorophyll maximum in the ocean.

## 1 Introduction

Convective mixing is one of the most important mechanisms supplying nutrients in the euphotic layer in mid- and high latitudes (see e.g. Sorokin, 2002; Williams & Follows, 2003, Severin et al., 2014). With the rise of stratification and irradiance vertically

entrained nutrients during winter are further consumed by phytoplankton, which causes the early-spring bloom in the upper layers of the ocean (Sverdrup, 1953; Sorokin, 2002). After the bloom, part of the nutrients in organic form sinks out to the nitroclyne and another part regenerates, which can fuel the phytoplankton bloom in the warm period of the year (Williams & Follows, 2003).



Particularly, in the Black Sea according to Lebedeva & Vostokov (1984), Karl & Knauer (1991) only a small fraction (~10%)
of particulate flux is exported to deep anoxic part of the sea, while the largest part of nutrients (90%) recirculates in the upper
layer (Oguz et al. 1999). The intensity of winter convection in the Black Sea, impact on the phytoplankton concentration in
both cold and warm period of the year, as well as on its yearly-averaged characteristics. The most intense winter-early spring
bloom of diatoms (Mashtakova, 1985; Sorokin, 2002; Mikaelyan et al., 2018) and the following early-summer bloom of
coccolithophores (Mikaelyan et al., 2015; Silkin et al., 2014, 2019) in the Black Sea are observed after cold winters and are
both related to the entrained in winter nutrients. The biomodelling study by Kubryakova et al. (2018) shows that the peak
values of the subsurface phytoplankton maximum in summer in the Black Sea also depend on the winter convection. Satellite
data demonstrated that the variability of surface concentration of chlorophyll-a (Chl) on interannual time scales is correlated
with winter sea surface temperature (Oguz et al., 2006; Finenko et al., 2014). Moreover, long-term analysis of in-situ data
(Mikaelyan et al., 2018) showed that winter temperature significantly affects the taxonomic composition and seasonal
succession of phytoplankton in the Black Sea throughout the whole warm period of the year.

A general feature of Chl vertical distribution in the Black Sea is the deepening of Chl peak during the warm period of a year
and a formation of a so-called deep chlorophyll-a maximum (DCM) (Sorokin, 1983; Vedernikov & Demidov, 1993), similarly
as in the other areas of the World Ocean at the same latitudes (see e.g. Mignot et al., 2014). According to the classical point of
view, phytoplankton occupies intermediate layers between the nutricline, situated in deeper layers, and illuminated layers near
the surface (Cullen, 2015). The nutrients entrainment in the euphotic layer in the warm period is strongly related to the short-
period mixing events (Williams & Follows, 2003) associated e.g. with storms (Iverson et al., 1974; Zhang et al., 2014;
Kubryakov, Zatsepin, et al., 2019), wind-driven or dynamic upwelling (McGillicuddy et al., 1998; Mikaelyan et al., 2020).
Such physical processes provide vertical fluxes of nutrients in the lower part of the euphotic zone, which is one of the important
reasons for the phytoplankton growth in its subsurface maximum in summer (Jonhson et al., 2010; Cullen, 2015).

Another reason for the formation of DCM is the summer rise of solar radiation, which leads to the widening of the euphotic
layer, and deepening of the productive zone (Letelier et al., 2004; Mignot et al., 2014; Lavigne et al., 2015). At the same time,
high values of irradiance may cause photoinhibition and decrease of Chl near the surface (Platt et al., 1982) caused by several
effects including non-photochemical quenching, photoinhibition, and photoadaptation (Falkowski & Raven, 2013). The latter
is partly associated with the increase of Chl content per cell (MacIntyre et al., 2002), documented for the Black Sea in (Finenko
et al., 2002, 2005; Churilova et al., 2019).

Intense phytoplankton bloom causes strong light attenuation, as photosynthetic pigments strongly absorb the light (Morel,
1991). This effect known as self-shading can significantly impact on the phytoplankton growth in deeper layers, where the
highest amount of nutrients is observed throughout the year (Shigesada & Okubo, 1981). The amount of Chl and related water
clarity largely control the depth of the euphotic zone (Shigesada & Okubo, 1981; Morel, 1991) and, therefore, impact on the
position of DCM (Letelier et al., 2004; Leach et al., 2018). The self-shading mechanism is one of the important factors driving
the phytoplankton variability and taxonomic composition in the most productive waters, e.g. lakes, rivers, and coastal waters
(Morel, 1991; Leach et al., 2018; Churilova et al., 2020). On seasonal time scales, Letelier et al. (2004) have shown that the





winter bloom of phytoplankton in the tropical Pacific Ocean leads to the additional shoaling of the euphotic zone, impacting on the development of phytoplankton in the deep layers. At the same time, the impact of self-shading on the interannual

variability of the vertical distribution of Chl in the open ocean and, particularly, in the central Black Sea is mostly overlooked. Recently-deployed Bio-Argo buoys give a possibility to obtain continuous measurements of Chl and light characteristics on the interannual time scales with high vertical and time resolution (Claustre et al., 2010; Mignot et al., 2014). In our study, we use the measurements of these buoys in the Black Sea to investigate the effect of winter mixing and irradiance on the vertical distribution of Chl in two years with contrasting winter conditions – warm in 2016 and cold in 2017.

The Black Sea bio-optical properties have several unique features, related mainly to the impact of large river inflow. Rivers cause its strong haline stratification and bring a vast amount of organic matter (Vladimirov et al., 1997; Mankovsky et al., 2010), which reduces the transparency of its water. As a result, the diffuse attenuation coefficient in the Black Sea is high (Organelli et al., 2017; Churilova et al., 2019) and the euphotic layer is only about 50 m (Vedernikov & Demidov, 1993). Therefore, the DCM in the Black Sea is relatively shallow occupying the upper 15-50 m. The variability of the thickness,

depth, and shape of summer DCM in the Black Sea was investigated in detail by Finenko et al. (2005), Krivenko (2010). Below the productive layer, the concentration of the nutrients began to rise sharply and the upper border of the nitroclyne in the Black Sea is situated approximately 50 m depth (Konovalov et al., 2005).

A strong vertical gradient of salinity suppresses the vertical mixing and the mixed layer depth (MLD) in the Black Sea in winter on average does not exceed 40-50 m. The spatial and temporal variability in winter MLD is strongly related to the

vertical displacement of the pycno-halocline (Kubryakov, Belokopytov, et al., 2019). Due to the strong haline stratification, the position of chemical layers in the Black Sea is tightly coupled to certain isopycnals and the variations of their concentration in density coordinates are significantly less than in $z$-coordinates (Konovalov et al., 2005; Tuğrul et al., 2014). The variations of the isopycnals are rather intense and are mainly caused by the dynamic forcing, which is represented by the large-scale cyclonic Rim Current and the abundance of mesoscale eddies (Korotaev et al., 2003). During winter convection, cold waters

do not penetrate through the pycno-halocline and form the cold intermediate layer with temperature (T<8°C) situated at 50-150 m depth, which further can be observed during the whole year (Staneva & Stanev, 1997; Korotaev et al., 2014). However, continuously observed warming causes the disappearance of the cold intermediate layer, which in the recent period is formed only in a very cold year, such as 2017 (Stanev et al., 2019).

Long-term rise of temperature observed globally (Behrenfeld et al., 2016) and in the Black Sea (Ginzburg et al., 2004;

Belokopytov, 2018; Stanev et al., 2019) increase the stratification in the upper layers, which may impact significantly on the convection mixing and related vertical nutrient fluxes. Such changes are expected to cause a decrease in overall biological productivity in the ocean (Behrenfeld et al., 2016). In our study, we show that the effect of self-shading may partly compensate for this decrease and will lead to the widening of the productive layer in the ocean.





## 2 Data and methods

The study is based on the bio-optical and hydrological measurements of Bio-Argo buoys in the deep part of the Black Sea in 2016 and 2017. These years have contrasting winter conditions with very cold winter documented in 2017 by (Stanev et al., 2019, Capet et al., 2020) and warm winter in 2016. We use the data of buoys #6901866 and #7900591, as they provide the simultaneous continuous measurements of both Chl and photosynthetic active radiation (PAR) in these years.

The measurements of both these buoys were made in the deep part of the Black Sea with depths of more than 1000 m. The

buoys were located in different areas of the basin (Fig. 1). Buoy #6901866 was moving in the cyclonic direction over the continental slope of the basin with the Rim Current. Buoy #7900591 was situated in the east-central part of the basin. As we will show below, despite these differences in the geographical position both buoys show similar results concerning the paper topic.

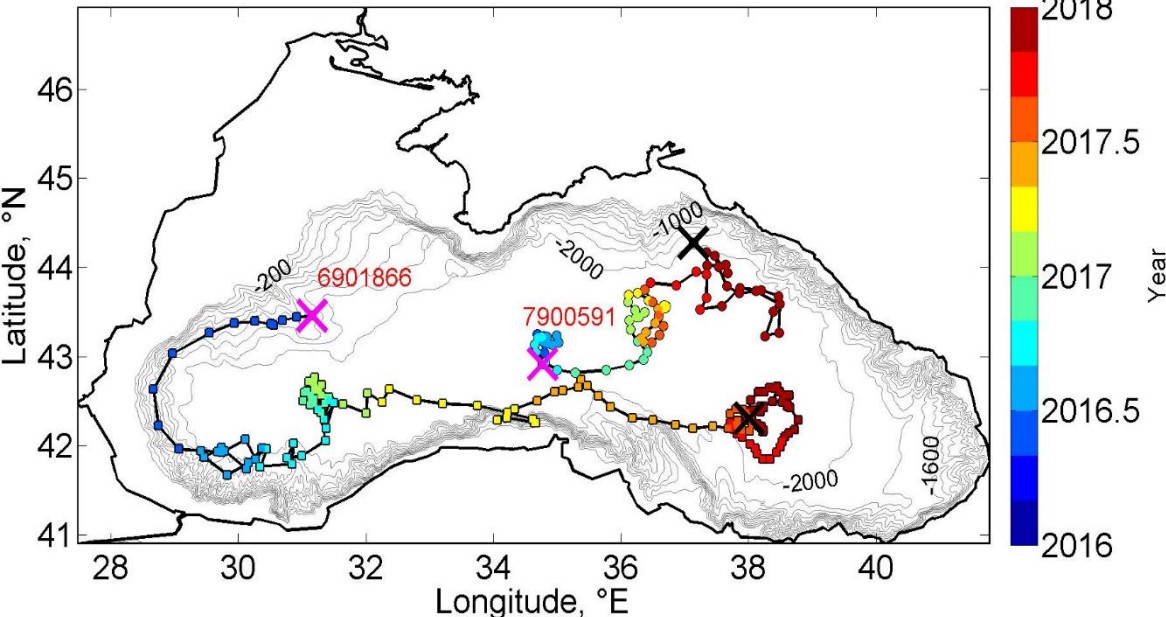

**Figure 1:** Trajectories of the buoys #6901866 and #7900591 in 2016 and 2017. The colorbar shows the date of the measurements. Purple crosses – the start of the trajectories, black crosses –end of trajectories. Gray lines show the position of isobaths.

Data on the Chl and downwelling instantaneous irradiance ($E_d$, μmol photons m$^{-2}$ s$^{-1}$) integrated over 400–700 nm were

analyzed. Chl (mg/m$^3$) was retrieved from the chlorophyll fluorescence sensor of the Wetlabs ECO Triplet Puck. The product "CHL_ADJUSTED" downloaded from ftp://ftp.ifremer.fr/ifremer/argo (Schmechtig et al., 2015) was used. This product includes the correction on non-photochemical quenching (Roesler et al., 2017) and the correction on the contribution of fluorescence by non-algal organic matter (Xing et al., 2017). As it is shown in (Xing et al., 2017li) the latter correction is very important for the Black Sea waters, which contains a huge amount of dissolved organic matter (Organelli et al., 2017), and the





adjusted product is about twice lower than non-adjusted. The description of the used protocol is given at http://www.argodatamgt.org/Documentation.

A multispectral ocean color radiometer (OCR-504, SATLANTIC Inc.) was used to measure the PAR in the water column. The precision of the device was 2.5 μmol photons/m²/s (https://www.seabird.com/ocr-504-irradiance-cosine-response-in-water/product-downloads?id=54627923874). Only $E_d$ measurements made near the local noontime (+/− 1 hour), which

represent approximately the maximal daily downwelling irradiance, were analyzed. Vertical gradients of $E_d$ were used to compute the diffuse attenuation coefficient $K_d(\lambda)$: $K_d(\lambda) = \ln\left(\frac{E_d(z+dz)}{E_d(z)}\right)/dz$, here z is depth, dz=1 m. All the profiles were visually quality controlled for consistency and absence of outliers.

The Bio-Argo buoys provided data on salinity and temperature at the time of the bio-optical measurements. All the data was visually quality-controlled for consistency with known T, S-distribution in the Black Sea. Eventually, we find no outliers in

the data of these buoys. MLD was estimated using a potential density threshold of 0.007 kg/m³. This criterion was justified by Kubryakov, Belokopytov, et al. (2019) on the base of the composite analysis of the long-term data for the Black Sea in 1985-2017.

Bio-Argo buoy data have a high time resolution (from 1 to 5 days) and vertical resolution (1 m), is regular and is publicly available at http://doi.org/10.17882/42182 or can be downloaded from the DAC (such as Coriolis).

Unfortunately, the optical-based nitrate measurements of Bio-Argo buoys in the Black Sea are poorly consistent with information on nitrates distribution known from numerous in-situ studies. Particularly Bio-Argo buoys show the persistent presence of more than 3 μM nitrates in the upper layer of the Black Sea throughout the year (see fig. S1), which is not consistent with 0.5 μM documented in many previous studies (Konovalov & Murray, 2001; Tuğrul et al., 2015). A possible reason for this is the complex optical characteristics of the Black Sea with a lot of dissolved organic matter, e.t.c. (see e.g. Organelli et

al., 2017). That is why this data was not used for this study.

## 3 Results

### 3.1 Physical characteristics in years with cold and warm winter

The results of this study are based on a comparative analysis of the physical and bio-optical characteristics of the Black Sea in 2016 and 2017, which were characterized by different winter thermal conditions. In 2016, the winter was significantly warmer

than in cold 2017 (Fig. 2a, b). The temperature in the upper 70-meter layer from January to March 2016 did not fall below 8.5°C (Fig. 2a), and its column-averaged value varied from 8.5 to 9.0°C (Fig. 3a, solid blue line). Due to such warm conditions in the winter of 2016, the cold intermediate layer with a temperature of less than 8°C was absent this year (Fig. 2a).

In 2017, the temperature in the upper 50-meter layer was on 1°C lower (Fig. 2b, Fig. 3a, blue dashed line) The winter of 2017 was one of the most severe in the recent period in the Black Sea. It was significantly colder than in warm 2016 and cause

significantly stronger vertical mixing than in 2016 (Stanev et al., 2019) reflected particularly in the rise of oxygen concentration





in the basin (Capet et al., 2020). The minimum temperature in February reached 7°C on the surface, and 7.5°C in the 0-70 m layer. As a result, the cold intermediate layer was formed in 2017 at 45-70 m and was persistently observed until September of 2017.

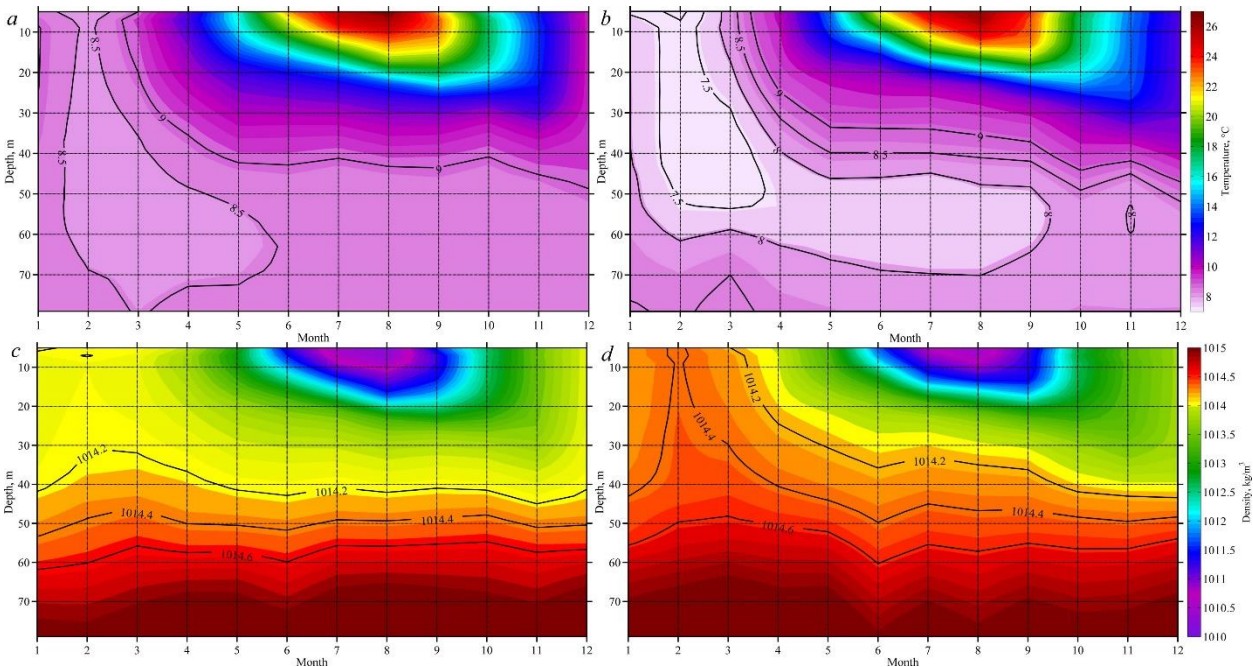

**Figure 2:** Monthly vertical diagram of temperature in 2016 **(a)** and 2017 **(b)**, potential density in 2016 **(c)** and 2017 **(d)**.

The decrease of temperature in 2017 led to an increase in the density of the surface layer (Fig. 2c, d) and intensification of conventional mixing. Maximum entrainment of deep isopycnals layers was observed in both years in February-March (Fig. 2c, d). In 2017, intensive cooling caused an outcropping of isopycnal 1014.4 kg/m$^3$ on the surface (Fig. 2d), while in 2016 the maximum density of the mixed layer did not exceed 1013.8 kg/m$^3$ (Fig. 2c). In the strongly stratified Black Sea, the location of nutricline is tightly coupled to certain isopycnals as it is shown in many chemical studies (Tuğrul et al., 1992; Konovalov et al., 2005). For example, the concentration of nitrates begins to gradually increase below the isopycnal of 1014 kg/m$^3$ (see Supplementary Fig. S1), and rises more sharply below the isopycnal of 1014.4 kg/m$^3$ where the upper part of the nutricline is located (Konovalov & Murray, 2001). The deeper isopycnals the winter convection reaches, the more new nutrients will be entrained in the upper layers. The tight relation between density and the position of chemical elements (see Konovalov et al., 2005) suggests that the density of the upper mixed layer can be used as a proxy for the qualitative estimation of the vertically entrained nutrients in winter (Kubryakova et al., 2018).



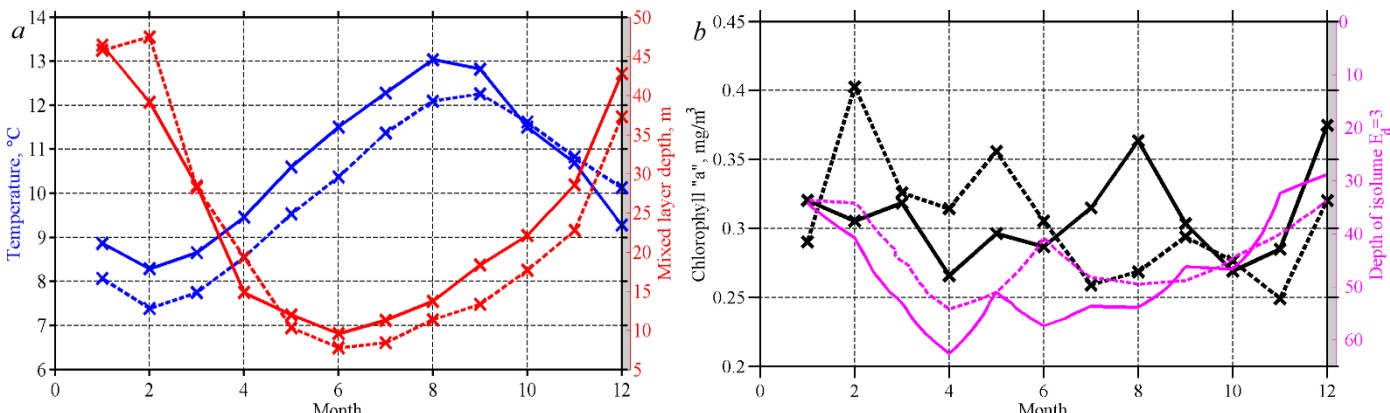

**Figure 3:** Monthly variability of column-averaged Chl in the 0-70 m layer (black color), column-averaged temperature (blue) in the 0-70 m layer, mixed layer depth (red), and depth of isolume $E_d=3$ μmol photons m$^{-2}$ s$^{-1}$ in 2016 (solid lines) and 2017 (dashed lines).

Due to stronger cooling, MLD (shown by the red line in Fig. 3a) in winter 2017 was also larger in 2016. However, this difference was rather small with MLD equal to 45 m in February 2017 compare to 40 m in 2016. The entrainment of deep isopycnals layers depends not only on the vertical mixing, but also on the vertical uplift of isopycnals during the seasonal winter intensification of cyclonic circulation in the Black Sea (Titov, 2004). Usually, the intensity of the large-scale circulation in the basin is observed in the cold years (Blatov et al., 1984; Oguz et al., 2006), as both of them are driven by the same atmospheric patterns – intensification of northeast winds from Eurasia (Kubryakov, Stanichny et al., 2019). The rise of cyclonic circulation and uplift of the pycno-halocline on the opposite decreases MLD (Kubryakov, Belokopytov, et al., 2019). That is why, in the Black Sea the MLD is not correlated with sea surface temperature (Titov, 2004), but strongly depends on dynamic forcing, such as eddies, large-scale circulation. Thereby, in the Black Sea density of the mixed layer rather than its depth reflect the intensity of the entrainment of new nutrients in the upper layers. In any way, in the considered case, both the depth and density of the upper mixed layer were larger in the cold winter of 2017.

To conclude, the above analysis is used to argue that the vertical entrainment of nutrients from deep isopycnals layers was more intense in the cold winter of 2017 than in the warm winter of 2016. The increase of nutrient concentration in the Black Sea in the cold years by in-situ data was previously documented by Tuğrul et al. (2015).

**3.2 Bio-optical characteristics in years with cold and warm winter**

The winter entrainment of nutrients in the upper layer caused the intense growth of phytoplankton, which is reflected in the increase in the Chl in February-March. The average monthly data of two buoys show that in February of cold 2017 Chl in the upper 30-meter layer reach 0.7 mg/m$^3$ (Fig. 4b) In the same period of 2016 Chl was only about 0.4-0.5 mg/m$^3$ (Fig. 4a). Column-averaged Chl in 0-70 m layer in 2017 was 0.4 mg/m$^3$ (Fig. 3b, black dashed line), which was on 0.1 mg/m$^3$ higher than in 2016 (Fig. 3b, black solid line).





**Figure 4:** Seasonal variability of Chl in 2016 **(a)** and 2017 **(b)**. The black line marks the border of the photic zone (maximum photosynthetic active radiation $E_d=3$ µmol photons m$^{-2}$ s$^{-1}$), the red line marks MLD. Seasonal variability of diffuse attenuation coefficient $K_d$ (m$^{-1}$) in 2016 **(c)**, 2017 **(d)**; seasonal variability of PAR in 2016 **(e)**, 2017 **(f)**.

Lower Chl values in 2016 in the upper 20-40 m layer were observed almost until the end of the year. From mid-March to July, the Chl in this layer in 2016 was about 0.3-0.45 mg/m$^3$, while in 2017 it was significantly higher – about 0.5-0.6 mg/m$^3$. The maximum difference was detected in March-June, when it reached 0.4 mg/m$^3$ in the 10-40 m layer.

These changes are especially well seen in the Fig. 5a, which shows the difference in Chl vertical distribution between 2017 and 2016. Almost throughout all year – from January to October – Chl in the upper 0-40 m layer was larger in cold 2017 than in warm 2016. This diagram evidences that the intense vertical convection in the cold winters led to an increase of biological





productivity not only during winter-spring bloom of phytoplankton, but also in the following months in agreement with the results of previous investigators (Burenkov et al., 2011; Finenko et al., 2014; Mikaelyan et al., 2015, 2018; Kubryakova et al., 2018). In particular, this effect is a probable reason for the unusually strong coccolithophore bloom in May-July 2017, which was documented on the base of satellite and Bio-Argo measurements by Kubryakov, Mikaelyan, et al. (2019).

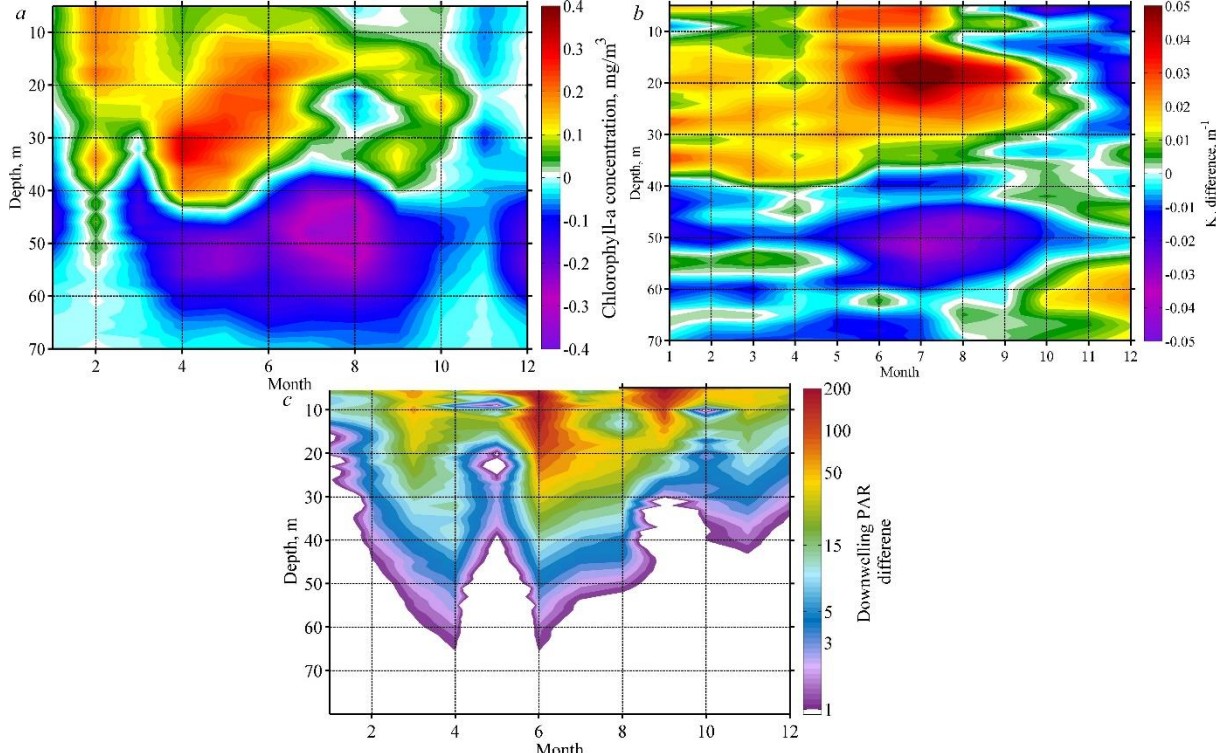


**Figure 5:** Seasonal diagram of a difference of vertical distribution of Chl **(a)**, $K_d$ **(b)** between 2017 and 2016, and PAR difference in 2017 and 2016 **(c)**.

In contrast, in deeper layers below 40 m depth Chl in cold 2017 was significantly lower than in warm 2016 (Fig. 5a). Starting 210 from March to November, the Chl values in the 40-60 m layer in warm 2016 were higher than in 2017 by 0.2-0.4 mg/m$^3$. The maximum excess of Chl in 2016 at these depths was observed in the summer period at 35-60 m, when its values were on 0.4-0.5 mg/m$^3$ more than in 2017.

Ten-daily diagram of Chl for individual Argo buoys #6901866 and #7900591 (Fig. 6a-d) demonstrate the same features in more detail. Both buoys show the presence of the intense short-period oscillations of Chl. Such local blooms in the warm 215 period of the year in the Black Sea are associated with the impact of dynamic forcing, such as storms and/or mesoscale eddies increasing the nutrient fluxes in the euphotic layer (Mikaelyan et al., 2017, 2020; Kubryakov, Zatsepin et al., 2019).





**Figure 6:** Time variability of Chl by the measurements of Bio-Argo buoy #7900591 for 2016 (**a**) and 2017 (**b**); buoy #6901866 for 2016 (**c**) and 2017 (**d**); the differences between 2017 and 2016 by buoy #7900591 (**e**) and buoy #6901866 (**f**).



At the same time, the most important for the task of the present study is that according to both buoys measurements Chl subsurface maximum in warm 2016 was deeper than in cold 2017. The diagram of the Chl difference in Fig. 6e, f clear shows these differences. In a 10-35 m layer, both buoys detected the rise of Chl in 2017 compared to 2016 on more than 0.2-0.5

mg/m$^3$. However, similar in magnitude negative anomalies were observed at deeper layers below 35 m. The strongest and widest negative anomalies were detected by buoy #6901866 over the continental slope in the whole 35-75 m in the summer period. In the central part of the basin buoy #7900591 detected such anomalies throughout the whole season in the narrower layer with a thickness of 20 m. Their position varied seasonally – deepen in summer to 40-60 m and uplift in autumn to 30-50 m, in agreement with the seasonal changes of PAR.

Chlorophyll-a is one of the main absorbing pigment, which significantly impact on the light penetration in the sea. Higher vertical entrainment of nutrients in the cold 2017 increased Chl in the upper layers, which lead to the rise of the light attenuation coefficient $K_d$ shown in Fig.4c, d. Coefficient $K_d$ in 2017 in the upper 30-meter layer from January to October exceeded 0.12 m$^{-1}$. The most prominent feature of $K_d$ distribution is the local maximum reaching values of 0.15 m$^{-1}$ at 10-30 m depth in Jule-September (Fig. 4d). This maximum was investigated in detail in (Kubryakov, Mikaelyan, et al., 2019), where it was related

to the release of dissolved organic matter (DOM) after enormously strong early-summer coccolithophore blooms formed due to intense winter convection in 2017. In comparison, in 2016, the values of $K_d$ in 0-40 m in February-October was only about 0.1 m$^{-1}$ (Fig. 4c). The diagram of $K_d$ difference shown in (Fig. 5b) is very similar to that for Chl (Fig. 5a) which supports the idea that the changes of $K_d$ are strongly related to the changes of chlorophyll-a distribution. Due to the higher amount of Chl and autochthonous DOM at the upper 0-30 m layer, $K_d$ in 2017 is higher on 0.01-0.05 m$^{-1}$ than in 2016. Smaller and opposite

in sign of negative differences (0.1-0.2 m$^{-1}$) are observed in deeper layers below 30 m, where $K_d$ is lower in 2017.

The rise of $K_d$ led to a decrease in PAR in the subsurface layers and shallowing of the euphotic zone. The euphotic zone is marked in Fig. 4a-d as the isolume Ed=3 µmol of photons m-2 s-1 (or 0.08 mmol photons m$^{-2}$ day$^{-1}$). This isolume was chosen because in the Black Sea the position of the upper border of relatively high Chl values follow this isolume, which indicate that this value of PAR can play a role of the compensational irradiance in the basin (Kubryakov et al., 2021 (manuscript in press)).

This isolume was located at 40-45 m in cold 2017, which was on more than 10 m shallower than in warm 2016, when it was located at 55-65 m (see their comparison in Fig. 3b – magenta lines). This indicates that in cold 2017, the growth of phytoplankton in the deep layers was limited by the low light conditions.

The decrease of Chl in the upper layers in warm 2016 led to a stronger penetration of PAR in the subsurface layers of the sea. The monthly diagram of the vertical distribution of PAR excess in 2016 compared to 2017 is shown in Fig. 5c. A significant

increase of PAR in warm 2016 by more than 1 µmol photons m$^{-2}$ s$^{-1}$ was observed in the entire 0-60 m layer throughout the whole year. In the 10-30 m layer, the values of PAR difference during the early spring bloom in February-April and autumn bloom in October-November reached 20 µmol photons m$^{-2}$ s$^{-1}$ (Fig. 5c). In summer, these differences were greatest and exceeded 200 µmol photons m$^{-2}$ s$^{-1}$ in the 0-25 m layer and were more than 10 µmol photons m$^{-2}$ s$^{-1}$ in the 25-40-m layer. Taking the value of compensational irradiance in the Black Sea as 3 µmol photons m$^{-2}$ s$^{-1}$, such a PAR increase caused a

significant widening of the euphotic zone.





As a result of PAR penetration, phytoplankton growth was increased by light in the deepest layers (40-65 m) below the maximum winter MLD (45 m). The penetration of light into this layer in warm 2016 due to the decrease of self-shading caused the growth of phytoplankton directly in the nitroclyne without mandatory mixing. A similar process is observed, in particular, in subtropical regions, where winter cooling is almost absent.

The above-discussed features of the differences in vertical Chl distribution in cold and warm year are well seen in its yearly-averaged profiles (Fig. 7). Data of both buoys show that the intense vertical entrainment of nutrients in the cold winter of 2017 lead to the sharper peak of Chl at 0-40 m layer, where yearly-averaged Chl was on 0.1 mg/m$^3$ (or about 20%) higher than in 2016. A similar increase in the summer peak of Chl in cold years were obtained in the modeling study of Kubryakova et al. (2018). At the same time in the warm 2016 with higher light penetration the subsurface maximum is noticeably wider. Values

of Chl higher than 0.2 mg/m$^3$ are observed at 0-60 m layer in 2016 compare to 0-45 m in 2017 (Fig. 7c). Due to this, the yearly-averaged Chl in warm 2016 at 40-60 m depth were on 0.1 mg/m$^3$ larger than in 2017.

   The integral of two curves are almost similar indicating that the same concentration of Chl formed in warm and cold year (Fig. 7c). Despite the significantly larger Chl in the upper 40-meter layer in 2017, the difference in column-averaged Chl over 0-70 m did not exceed 0.1 mg/m$^3$ in the first half of the year, except the period of intense convection in February, when it reaches

0.15 mg/m$^3$ (Fig. 3b). In summer, the sign of this difference changed to the opposite. Due to the strong rise of light attenuation, the summer maximum of Chl was suppressed in 2017. In 2016, light penetrated to the upper layer of nitroclyne, which causes the development of a deep and wide subsurface maximum in July-August 2016 at 20-60 m (Fig. 4a, Fig. 6a, c). At this time the column-averaged Chl in 2016 exceeded its values in 2017 by the same value of 0.1 mg/m$^3$ (Fig. 3b). Thus, the decrease in Chl in the first half of the year in the upper layer of 2016 was compensated by its increase in the summer period due to the

lack of self-shading effect and the development of a deep Chl maximum directly in the nitroclyne. Because of this, the yearly-averaged integral Chl in the euphotic layer in 2016 and 2017 became comparable (Fig. 3b, Fig.7).

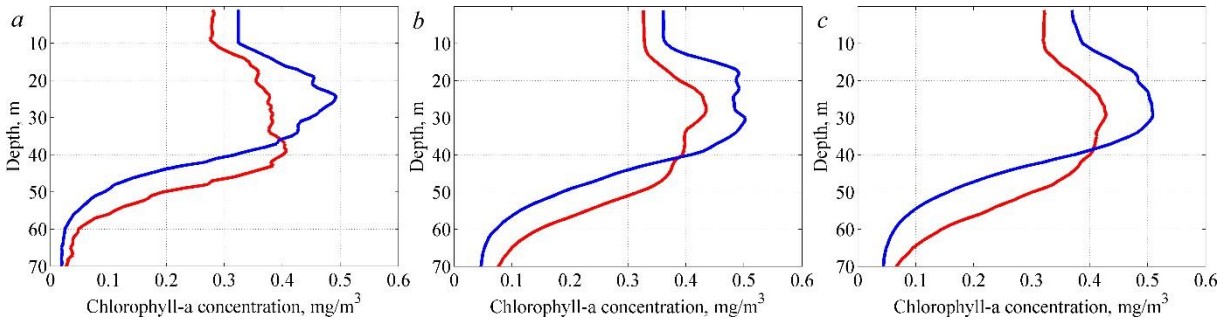

**Figure 7:** Average profile of Chl in 2016 and 2017 by the measurements of the buoy #7900591 (**a**) and buoy #6901866 (**b**)
and both buoy measurements (**c**).



## 4 Discussion

The reasons for the variability of the characteristics of DCM is one of the important and actively investigated oceanographic tasks (Cullen, 2015; Leach et al., 2018; Barbieux et al., 2019). Variability of its position and strength are related in different studies to the vertical distribution of nutrients (Hartman et al., 2014; Barbieux et al., 2019), optical characteristics of water and light availability (Morel, 1991; Mignot et al., 2014; Leach et al., 2018), density stratification (Navarro & Ruiz, 2013). Our study approves that all these factors are important and describe a relation between their impact on the vertical distribution of Chl on the example of the Black Sea (see scheme in Fig. 8). Nutricline in the highly-stratified waters of the Black Sea is closely related to the isopycnals position (Konovalov et al., 2005). The increase of density of the upper mixed layer in winter leads to the convective mixing reaching the isopycnals with the same density. A deep isopycnal layer with a high amount of nutrients mixes with surface waters and defines the concentration of nutrients in the winter mixed layer (0-40 m in the Black Sea) (Kubryakova et al., 2018). Further thermal stratification stabilizes the water column. Entrained in winter period nutrients and the rise of the irradiance causes the following spring growth of phytoplankton. Part of them regenerates and another part sinks out to the nitroclyne. The regenerated nutrients further are used by the summer population of phytoplankton. For example, in the Black Sea, the strong coccolithophore bloom emerges after strong spring bloom of diatoms and their magnitude depends on the winter sea surface temperature (Burenkov, 2011; Silkin et al., 2014; Mikaelyan et al., 2015, 2018). Thus, the phytoplankton concentration in the upper layer through the whole year largely depends on the density of the preceding winter mixed layer, which is approved by the difference in Chl distribution in warm and cold year in Fig. 4a, b, 5a, 6e, f.

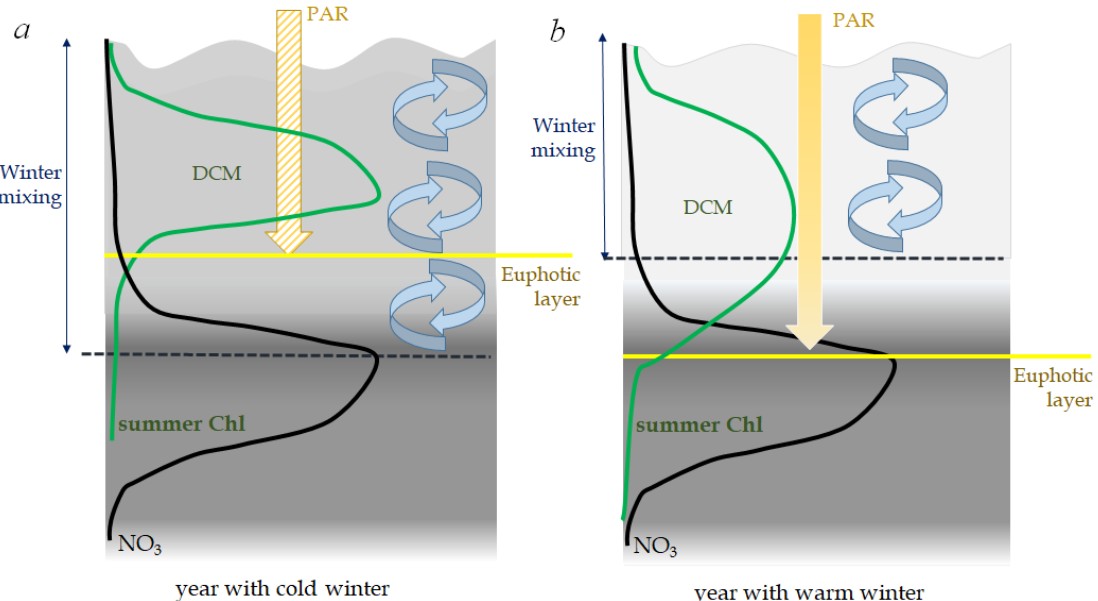

**Figure 8:** A scheme of the impact of convection and self-shading on the vertical distribution of chlorophyll-a. **(a)** In a year with cold winter, the larger amount of nutrients (grey color) is convectively entrained in the upper layer, which increases the





growth of phytoplankton in the upper layer and causes self-shading of the deeper layer. Therefore, DCM moves to the upper layer. **(b)** In years with warm winter convective nutrient fluxes are low, the amount of phytoplankton and light attenuation decreases. In the summer period with the increase of PAR, light penetrates the upper layer of nitrocline and causes intense and deep summer subsurface bloom. Therefore, the total amount of nutrients used by the phytoplankton in both years is comparable.

In the warm period of a year PAR strongly increases, the euphotic layer deepens, and the surface layer becomes over-illuminated, which is one of the possible reasons for the formation of DCM in summer (Platt et al., 1982; Mignot et al., 2014). The phytoplankton pigments, absorb light, and its variability largely defines the optical characteristics of the water through the self-shading mechanism (Morel, 1991; Churilova et al., 2020). Thereby in the water with higher nutrient concentration (e.g. after cold winter), the light penetration to the deeper layer weakens (Fig. 5b, c). The water clarity decreases and DCM is situated closer to the illuminated layers (Fig. 8a) near the surface (Mignot et al., 2014; Leach et al., 2018), as we observed in 2017 in the Black Sea (Fig. 7). On the opposite, in the water with relatively low winter nutrient fluxes (as in 2016 in the Black Sea), Chl and biomass of phytoplankton decreases. The absence of self-shading promotes the light penetration to the deeper layer, which leads to the deepening and widening of the DCM (Fig. 7, Fig. 8b). These arguments suggest that DCM displacement is driven by the amount of nutrients and related self-shading by formed phytoplankton.

Light attenuation depends not only on the phytoplankton or Chl, but also on the attenuation by clear water and DOM (Morel, 1991), which concentration in the Black Sea is very high due to the intense river discharge (Mankovsky et al., 2010; Organelli et al., 2017). Another strong source of light attenuation, which is nutrient-dependent, is autochthonous DOM formed due to the release of lipids during lysis of phytoplankton cells. Kubryakov, Mikaeyan et al. (2019) based on the Bio-Argo data on diffuse attenuation at 390 nm and backscattering measurement show that one of the most significant sources of DOM is related to the lysis of coccolithophore cells during the termination of their early-summer bloom possibly mediated by a viral attack. As it is shown by Burenkov et al. (2011), Mikaelyan et al. (2011, 2015) the intensity of coccolithophore blooms in the Black Sea is significantly related to winter temperature and the amount of entrained nutrients. Particularly, the extremely strong coccolithophore bloom was observed in the Black Sea after cold 2017, which results in the observed maximum light attenuation in July-August in 2017 at 10-30 m depth (Kubryakov, Mikaeyan et al., 2019). Such release of DOM caused by strong summer coccolithophore blooms or blooms, as well as other types of the phytoplankton in cold years additionally reduces the water transparency. This effect will additionally decrease the light penetration in cold years with a larger amount of nutrients. Therefore, the release of light-absorbing autochthonous DOM plays an additional role in equalizing the total productivity in warm and cold years.

It is also important that in the year with weak winter nutrient fluxes during summer seasonal maximum of irradiance light penetrates to the most deeper layers, where nutrient concentration is high. Particularly, in the considered case in warm 2016 PAR reached 40-60 m depth, where the upper border of nitrocline is located in the Black Sea. At these depths, the concentration of nitrates sharply increases (Supplementary Fig. S1). At the same time, the winter mixed layer in the Black Sea is only about 40-45 m (Titov, 2004; Kubryakov, Belokopytov, et al., 2019) (Fig. 3a). Underlying rich in the nutrients layers were unaffected





by winter dilution with poor in nutrients surface waters. The presence of light and nutrients cause the growth of phytoplankton. The maximum of Chl in this case forms closer to the layers with a high concentration of nutrients. Such a position should increase the efficiency of the upward nutrient fluxes, which are proportional to the vertical gradients of nutrient concentrations.

This process is most important in the warm period of a year, when phytoplankton is fueled mainly from below by the intense diapycnal mixing caused by storms (Iverson et al., 1974; Zhang et al., 2014; Chacko, 2017) or, e.g. eddy-driven upwelling (McGillicuddy et al., 1998; Mikaelyan et al., 2020). Particularly, in the Black Sea, several storms in August 2015 led to the intense subsurface maximum of Chl at the base of the euphotic layer reaching (Kubryakov, Zatsepin, et al., 2019). The closeness of DCM and nutricline increases the nutrient fluxes in the summer of the years with warm winter, which compensates

for their decrease in winter caused by weak convective entrainment.

   The proposed mechanism can help to explain the observed changes in seasonal succession of phytoplankton after cold and usual winters, studied in detail by Mikaelyan et al. (2018). Particularly Mikaelyan et al. (2018) show that after cold winters in the Black Sea the biomass of diatoms is high in spring and low in summer. The opposite is observed after warm winters.

   Large nutrient fluxes in cold winter cause intense spring diatom bloom (Mashtakova, 1985; Sorokin, 2002; Silkin et al., 2014)

which is followed by strong coccolithophore bloom (Mikaelyan et al., 2015). The lysis of coccolithophores causes the intense rise of DOM in the upper layer (Kubryakov, Mikaelyan, et al., 2019). Light attenuation decreases the thickness of the euphotic zone and the efficiency of the upward nutrient fluxes in it from the nutricline. The release of DOM also can trigger the microbial loop, which transfers the trophic energy in smaller species. Both these factors can explain the high biomass of diatoms in spring and low in summer after cold winters. In contrast, after warm winter phytoplankton in spring grows in low-nutrient

conditions. Thereby in spring diatom biomass is lower and dinoflagellates biomass reaches maximum, according to (Mikaelyan et al., 2018). However, in summer phytoplankton develops in the deeper layer with a higher amount of nutrients, which may explain the higher diatom biomass and diversity in warmer years documented by Mikaelyan et al. (2018).

   It should be noted, the phytoplankton after cold and warm winter grows at different depths and, therefore, the environmental (light and nutrients) conditions change significantly. Deepening of the euphotic layer may also promote the growth of species

adapted to low light with low biomass and high Chl content in cells (Falkowski & La Roshe, 1991; MacIntyre et al., 2002; Latasa et al., 2017). Particularly, in the Black Sea, the lower border of the euphotic zone is characterized by the domination of small flagellates and unicellular cyanobacteria (Churilova et al., 2019; Mikaelyan et al., 2020), which may have an advantage in the years with warm winters.

   Globally-observed warming and the rise of thermal stratification are expected to decrease the vertical nutrient fluxes associated

with winter cooling and primary production in the ocean (Behrenfeld et al., 2016). Particularly, in the Black Sea the long-term rise of sea surface temperature on about 1.5℃ per 10 years was documented by Ginzburg et al. (2004), Oguz et al. (2006) and already lead to the disappearance of the cold intermediate layer in recent years (Belokopytov & Shokurova, 2005; Knysh et al., 2011; Piotuch et al., 2011; Capet et al., 2020). The result of the presented manuscript suggests that the continuous weakening of thermal convection and associated weakening of the nutrient fluxes will also cause a decrease of self-shading by

Chl and an increase of the water clarity. Such changes in water optical properties may cause the long-term deepening of DCM





in the ocean to the rich in nutrients deeper layers. We may suggest that, at first, such displacement may enhance the production in a warm period of the year during the seasonal peak of irradiance and compensate for the decrease of productivity in winter-spring. However, with the deepening of DCM, remineralization of the organic matter and settling of the particles will occur also in deeper layers. These changes will cause the consequent deepening of the nutricline. In such a way the ecosystem of the

temperate Black Sea will transit to a tropical state, with very deep nitroclyne and DCM. This trend is especially dangerous in the oxygen minimum zones, such as observed in the Black Sea, where nitrates are effectively removed at the suboxic interface due to the process of denitrification, the oxidation of nitrates by particulate manganese, and others (Konovalov et al., 2008). In such areas, the intense removal of nitrates from below may cause the thinning of the nutricline, the additional decrease of biological productivity, and oxygen production. Such a process also can promote the changes in the taxonomic composition

of phytoplankton to low-light adapted species.

## 5 Conclusions

A comparative analysis of the bio-optical characteristics of the Black Sea for 2016 and 2017 based on Bio-Argo buoys showed that the intensity of winter convection largely controls the intensity and the position of the subsurface maximum Chl throughout the year:

- intense convection in the cold winter of 2017 caused the entrainment of nutrient-rich waters from deep isopycnals layers and a significant increase of Chl in the upper layers. In the 0-40 m layer from March to October, the Chl values exceeded that in warmer 2016 by 0.2-0.4 mg/m$^3$. This increase in Chl and related autochthonous DOM led to the rise of light attenuation and the decrease of Chl in the deeper layers due to low light conditions. As a result, PAR and Chl in the deepest 40-60 m layer in 2017 were significantly lower than in the warm 2016;

- the weakening of vertical fluxes of nutrients in the warm winter of 2016 caused a significant decrease of Chl in the upper 0-40 m layer compare to 2017. This increases the transparency of the water, and PAR penetrated to a depth of 40-60 m in the upper layer of nitroclyne, which was unaffected by winter mixing with winter MLD about 40 m. As a result, in the lower 40-60 m layer, Chl in 2016 exceeded its values in 2017 by 0.2-0.4 mg/m$^3$. The maximum rise of Chl in 2016 was observed in summer during the peak of irradiance, which penetrated to the deepest layers of the Black Sea.

Low winter nutrient fluxes in warm winters decrease the productivity in upper layers, which promote the light penetration in summer to the upper layers of nitroclyne and intensify summer nutrient fluxes from below. As a result, the obtained estimates of the integral layer of Chl were comparable in years with warm and cold winters. At the same time, the intensity of winter convection led to significant changes in the vertical distribution of Chl. In the cold years, DCM was relatively sharp and located in the upper layers, while in the warm years DCM widens significantly, but Chl in it was less and distributed more uniformly.

The observed inverse relation between winter and summer nutrient fluxes can be one of the important reasons for the observed interannual changes of seasonal succession in the ocean.





**Data availability**

Bio-Argo data were made freely available from IFREMER data archive [ftp://ftp.ifremer.fr/].

**Author contribution**

Elena Kubryakova and Arseny Kubryakov processed the data of Bio-Argo buoys, performed the analysis. Elena Kubryakova drafted the manuscript and designed the figures. Arseny Kubryakov was involved in planning and supervised the work.

**Competing interests**

The authors declare that they have no known competing financial interests or personal relationships that could have appeared to influence the work reported in this paper.

**Acknowledgments**


Analysis of the chlorophyll-a and light attenuation variability from Bio-Argo buoys is supported by the Russian Science Foundation (grant No. 19-77-00029). The study of the impact of PAR on the position of DCM is supported by the Russian Foundation for Basic Research (grant No. 20-05-00068). Data acquisition and processing were made with the support of Marine Hydrophysical Institute, Russian Academy of Sciences, under state assignment No. 0827-2019-0002.

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
