# Peer review of "Warmer winters causes an increase of chlorophyll-a concentration in deeper layers: the opposite role of convection and self-shading on the example of the Black Sea"

_Biogeosciences, 2020_

## Referee Comment (RC1) · Nicolas Mayot (Referee) · 10 Nov 2020

Kubryakova and Kubryakov analyzed in situ data from two biogeochemical-Argo floats deployed in the Black Sea. They were interested in how the winter season could influence the phytoplankton biomass (vertical distribution and concentration) for the rest of the year. For this, they focused on observed differences between two years: 2016 and 2017.

Their manuscript is based on the spatiotemporal variabilities of several measured pa-

rameters: temperature, salinity, chlorophyll-a concentration and downwelling irradiance. In 2016, the winter season was warmer and associated with a lower vertical mixing than in 2017. The phytoplankton spring bloom intensity was lower in 2016 (lower chlorophyll-a concentration) than in 2017. In summer 2016, the Deep Chlorophyll Maximum (DCM) was deeper than in 2017. Authors suggested that in the Black Sea, a warm winter reduces the nutrient availability for the whole year, which decreases the phytoplankton spring bloom intensity and reduces the light attenuation in summer because of lower concentrations in phytoplankton cells and dissolved organic matter.

Such detailed study of physical-biogeochemical variables in the Black Sea based on in situ data is original and could greatly contribute to increase our understanding of physical-biological processes in this basin. However, some authors' assumptions are questionable and I have several specific comments. Therefore, the paper will likely be a significant scientific contribution with major revisions.

Major comments:

1) Authors wrote several times in the manuscript that in the Black Sea, as in other oligotrophic basins, the nitracline is closely connected to an isopycnal (for examples, see lines 80, 156, 161 and 287). When looking at figure 2 c-d, we can clearly see that isopynals between 1014 kg/m3 and 1014.4 kg/m3 (associated with the nitracline, see line 158 and figure S1) are shallower in summer 2017 than in summer 2016. Therefore, in summer 2017 the nitracline (or nutricline) is shallower than in summer 2016. The low chlorophyll-a concentration in 2017 between 40-60 m is probably due to a change in in the depth of isopycnals and not to a "rise of light attenuation", as argued by authors (line 387).

Line 343 – "The closeness of DCM and nutricline increases the nutrient fluxes in the summer of the years with warm winter, which compensates for their decrease in winter caused by weak convective entrainment". Such statement is false, because authors did not discuss differences in the depth of the nutricline between summer 2016 and

summer 2017.

In addition, when looking at figures 4a-b, chlorophyll-a concentration at the DCM is higher in 2017 than in 2016. Considering that isopycnals were shallower in 2017 than in 2016, and based on figures 4e-f, one could argue that the light level at the nutricline is higher in 2017 than in 2016. A higher light level at the nutricline could induce a higher phytoplankton production and biomass in the DCM, as observed in 2017. Such mechanism, clearly explained in the comprehensive review of Cullen (2015), is crucial and need to be discussed.

2) When studying DCM, it is important to know if the DCM is also a subsurface phytoplankton biomass maximum. Because, changes in chlorophyll-a concentration at a DCM could be induced by variations in intracellular chlorophyll-a concentration or by changes in phytoplankton biomass. In the current version of the manuscript, this point is not discussed.

When using data from biogeochemical-Argo floats equipped with an ECO Triplet (as here, line 110), measurements of particulate backscattering coefficient (bbp), a proxy of the particulate organic carbon, are available. These measurements have been used in a paper published by one of the authors (Kubryako et al., 2019). The manuscript will be more comprehensive if those measurements could be added to the current analysis.

Minor comments:

Line 11 – "caused an increase of Chl in winter up to 0.6-0.7 mg/m3 compared to a warm winter of 2016": It is unclear. There is an increase of Chl in winter in both years.

Line 19 – "more productive". That is a suggestion. Chlorophyll-a concentration measurements are not primary production estimates.

Line 24 – "With the rise of stratification and irradiance vertically. . ." A comma is needed after the word "irradiance".

Line 26 – "After the bloom, part of the nutrients. . .fuel the phytoplankton bloom". How

many blooms are there in a year?

Line 27 – "nitroclyne". It is a wrong spelling, used several times in the manuscript. Need to be replaced everywhere by "nitracline" or "nutricline". However, here it could be replaced by "euphotic zone".

Line 31 – "...convection in the Black Sea, impact on the...". Need to be replaced by "...convection in the Black Sea impacts on the..."

Line 35 – replace "biomodelling" by "modelling"

Line 45 – The diapycnal diffusivity could be an important physical process for nutrient supply. Maybe it is the most important in summer.

Line 52 – The relationship between expressions: "high values of irradiance may cause photoinhibition", "photoadaptation" and "increase of Chl content per cell" is unclear.

Line 66 – Replace "Bio-Argo buoys" by "biogeochemical-Argo floats". Throughout the manuscript, the word "buoys" should be replaced by "floats".

Line 89 – "Long-term rise of temperature observed globally (Behrenfeld et al., 2016)" This study did not focus on the long-term rise of temperature.

Line 121 – "ðİŘ¿ðİŚŚ $(\lambda)$" The letter z is missing in the parentheses.

Line 124 – All measurements from Argo floats have quality control flag values. Did authors check them?

Line 125 – Information about the process to obtain potential density values from in situ temperature and salinity measurements are missing.

Figure 2 – Is it in situ or potential temperature? Potential density? The colorscale or isolines are inconsistent. For example, in panel C, the isopycnal 1014.2 has a yellow background and in the panel D has an orange background.

Here and throughout the manuscript, could the authors use another colormap, not a

rainbow one: "In the 2007 IEEE article, Rainbow Color Map (Still) Considered Harmful, authors David Borland and Russell M. Taylor II from the University of North Carolina at Chapel Hill stated, "The rainbow color map confuses viewers through its lack of perceptual ordering, obscures data through its uncontrolled luminance variation, and actively misleads interpretation through the introduction of non-data-dependent gradients.""

Line 154 – "conventional mixing", is it the right term?

Section 3.1, Figure 2 and Figure 3 – Regarding physical parameters, there are no information or comments about potential differences between the two floats for a same year due to differences in their geographical locations. Are there any differences in the vertical distribution of temperature, isopycnals and MLD between the two floats for a same year? In addition, time series of temperature and density for each float should be added in supplementary materials. The sentence, line 101: "As we will show below, despite these differences in the geographical position both buoys show similar results concerning the paper topic" needs to be supported by figures and values for all physical and bio-optical parameters used here.

Line 201 – "productivity" chlorophyll-a concentration is a proxy for phytoplankton biomass not phytoplankton production.

Line 256 to 259 – This is a suggestion, it should be discussed in the discussion section.

Figure 7 – Why yearly-averaged profiles and not only summer (and winter) profiles.

Line 267 and figure 3b – It would be better to describe time series of depth integrated chlorophyll-a concentration and not column-averaged.

Line 274 to 276 – This is a suggestion, it should be discussed in the discussion section.

Line 308 – "the euphotic layer deepens, and the surface layer becomes over-illuminated" It would be better to use the euphotic depth definition. What is the meaning of "over-illuminated"?

---

## Short Comment (SC2) · 24 Nov 2020

Vladimir Silkin (Referee) vsilkin@mail.ru Understanding the critical role of winter convective mixing in the annual dynamics of phytoplankton is well known. However, this factor's effect on the vertical distribution of chlorophyll remained unclear, which was due to the lack of reliable methods. The development and improvement of methods for remote sensing of the sea surface have made it possible to obtain information about the environmental situation over large areas. However, this information is only concerned with the sea surface. The emergence of floating buoy technology allows solving the

vertical distribution of the environmental indicators. The authors of this paper used Biogeochemical-Argo floats' technology to find out the regularities of the vertical distribution of chlorophyll and estimate the depth maximum of chlorophyll (DCM) in the Black sea. Comparing the warm year 2016 with relatively weak convective mixing and the cold year 2017 with intense mixing, they found significant differences in the DCM position. To explain this phenomenon, the authors propose the hypothesis of regulating the DCM position using light. It is based on the idea that chlorophyll concentration in UML depends on the amount of nitrogen received in winter. Since the amount of incoming nitrogen is determined by the intensity of convective mixing in winter, after cold winters, should be expected higher chlorophyll concentrations in UML and, therefore, higher light absorption. By studying the light attenuation coefficient's distribution with depth, the authors revealed the critical role of the DCM position's self-shadowing effect. My familiarity with the previous version (dated June 16, 2020) and the current version (dated October 15), as well as with the comments of reviewers, allowed me to conclude that the new version meets all the requirements and can be published. However, it seems to me that the paper will be improved if the authors take into account my comments and recommendations: 1. In the Introduction, it is necessary to formulate the research objectives more clearly. In particular, the sentence on line 68 should be moved to the end of the Introduction. 2. On lines 78 - please provide a reference. 3. In the Results, provide only the authors' results and move the comments and reference to the Discussion (Lines, 144-145, 157-158, 160-163, 173-183, 201-205, 2016, 234, 244, 265). Individual comments and recommendations: 1. Line 19 and 201-chlorophyll concentration and productivity are not the same things. 2. Line 35 - "the biomodelling study" rewrite as "the modeling study.Ż 3. Line 85 - Cold intermediate layer mark as CIL and use it later in the article. 4. Lines 50-55. Rewrite these sentences "At the same time, high values of irradiance may cause photoinhibition and decrease of Chl near the surface (Platt et al., 1982) caused by several effects including non-photochemical quenching, photoinhibition, and photoadaptation (Falkowski & Raven, 2013). The latter is partly associated with the increase of Chl content per cell

(MacIntyre et al., 2002), documented for the Black Sea in (Finenko et al., 2002, 2005; Churilova et al., 2019)Âż. Here it would be best if you replaced photoadaptation with photoacclimation, since changes in the chlorophyll content in the cell are acclimation. Further, the chlorophyll content in the cell increases when the light intensity decreases. Therefore, it decreases at the surface. 5. Line 110-111. Rewrite this sentence. 6. Line 121. More correct to write Kd(z) rather than Kd($\lambda$) since this parameter changes with depth. 7. Line 129-decode DAC. 8. Line 187-there is no dot at the end of the sentence. 9. Line 315-chlorophyll and biomass are not the same things. 10. Figure 4 is missing the dimension for PAR. 11. In the caption to Figure 7, there is no designation of the curves (red and blue).

---

## Referee Comment (RC2) · Anonymous Referee #2 · 11 Dec 2020

**General Comments**

The authors investigate the drivers of differences in the vertical distribution of chlorophyll-a between 2016 and 2017 in the Black Sea using BGC-ARGO data. A key feature of interest in the vertical distribution is the so-called deep chlorophyll maximum (DCM), which the authors show is deeper and less intense in 2016 than in 2017. They account for this difference by arguing that cold atmospheric conditions in the winter of 2017 led to convective mixing and nutrient entrainment, thus increasing winter production. It is then argued that this increased production led to enhanced self shading in 2017, which accounts for why the DCM is shallower compared to 2016.

In general, I agree with other reviewers that the hypothesis presented is interesting and could represent a significant contribution to the question of what factors control the DCM. However, I also agree that currently the authors do not present sufficient evidence to support their hypothesis. Furthermore, the methodology requires some important revisions which I explain below. I therefore recommend that the following revisions be undertaken prior to publication:

1. All monthly averaging should be removed or only added to supplement the higher frequency data. This is actually why there is little difference seen in the MLD between the 2 years - the differences have been averaged out. Below I show an example of temperature profiles for early February comparing the 2 years. Here it is clear that the MLD is deeper in 2017 by ~20 m, although if you average over the whole month you won't see much difference.

[Figure]

This highlights that the phenomenon being investigated occurs at much higher frequency than monthly, which needs to be taken into account in more detail than is currently done.

2. Similarly to point 1 above, the data should be presented with as little interpolation as possible. It is clear from figures 2, 4, 5 and 6 that some kind of spatiotemporal interpolation has been done to produce such highly "smoothed" plots. Below I show an example of how the chl-a data look for float 6901866 with a minimal amount of interpolation (here I only use a linear interpolation in the "depth"

dimension for the missing data, and gaps of greater than 5 m are not interpolated) I suggest to change the figures to something more like this, which portrays the data more accurately:

[Figure]

Here it is clear that the high chl-a values seen in winter of 2017 are actually composed of 2 short periods (10-15 days) of elevated growth, one in December and another stronger one in March. Figure 2 in the current manuscript makes it seem like one long period of sustained growth. Figure 6 does actually show these 2 pulses, but since 2016 and 2017 are split into separate panels one cannot easily see the 2 distinct growth periods. The plot above also shows that the DCM is most intense (highest chl-a) in the autumn of 2018 - it might be interesting to look into why this is the case.

3. I follow the argument that the upliftment of isopycnals is associated with a rise in the nutricline and therefore nutrient entrainment into the MLD. However, I would argue that simply referring to other literature where this relationship has been established is not sufficient to say that it has occurred in the present case. Since

this entrainment of nutrients is key to the argument being made, it follows that it should be explicitly shown with data. Here I recognise that the nitrate data may be biased in these particular floats as the authors have suggested. However, the important point is that nitrate concentrations should be higher in the cold 2017 year, so biases in the concentration may not preclude the use of this data (since we look for relative differences, not absolute values). So long as the bias is properly taken into account I would argue that the data should be used to support the argument.

If the data are really not appropriate, perhaps other proxies for entrainment of deep water could be used (e.g. dissolved oxygen)?

4. If convective mixing is indeed present in winter of 2017, then one should be able to see strong cooling events preceding the mixing events. For this one could perhaps use a reanalysis product or something similar. The heat flux could even be estimated for these cooling events, although it may be enough to just correlate temperature anomalies with the mixing events. If there are indeed strong cooling events preceding the mixing, then this would certainly strengthen the argument.

5. I recommend that the authors provide a quantitative estimate of the DCM depth, so that its temporal variability be assessed objectively. I can think of various ways this could be achieved, perhaps by obtaining the mean depth of the 90th or 95th percentile of chl-a concentration for each profile. A time series of the DCM depth could then be produced for both floats and the cold/warm years compared quantitatively.

6. The level of English in some parts of the manuscript detracts from the value of the science being presented. I provide some suggestions for specific passages below, however, I would strongly suggest that the authors further edit the manuscript to improve clarity and the communication of the findings.

**Specific Comments**

All figures: The captions lack detail and in many cases are unclear. I suggest carefully reviewing them, adding additional details and rewording to avoid confusion. I give some examples below, but I suggest to revise all captions.

**Introduction**

1. Line 27 (and subsequent use): I'm not sure what is meant by " nitroclyne." Please define this.

2. Lines 58-59. Is this really true that: " The amount of Chl and related water clarity largely control the depth of the euphotic zone (Shigesada & Okubo, 1981; Morel, 1991) ." What about solar angle, time of year? Non-organic particles? Time of year is mentioned earlier in the text, but here it seems like Chl is essentially the only factor. I would reword to " The amount of Chl and related water clarity *strongly impact* the depth of the euphotic zone … "

3. Line 47. What is meant by the term " dynamic upwelling"? Please clarify in the text or reword, since this is not standard terminology.

4. Line 62 -63. What is the degree of shoaling of the euphotic zone reported in Letelier et al. (2004)? How is phytoplankton impacted and what is specifically meant by "deep layers" (i.e. how deep)?

5. Lines 80-82. *"Due to the strong haline stratification, the position of chemical layers in the Black Sea is tightly coupled to certain isopycnals and the variations of their concentration in density coordinates are significantly less than in z-coordinates.* " Do you mean that vertical variations in the concentration of certain chemicals is significantly less in density coordinates than in z-coordinates? If so, please state this more clearly since the wording is potentially ambiguous. I would also suggest briefly stating why this is important/ significant.

6. Lines 173 - 179: Do you mean here that large-scale circulation is intensified in cold years? If so, a revision of the wording is needed to make this clear. In addition, you would need to describe this phenomenon in more detail (i.e. what is the mechanism?).

**Results**

7. Lines 223 - 229: This passage is currently very unclear. What negative anomalies are the authors referring to? Do they mean the negative values shown in Figure 6e and f? In that case, they should not be referred to as anomalies (which suggest a difference with respect to a long term mean) but as differences

(higher or lower chl-a in 2017/2016) or perhaps just "negative values." I would suggest revising these lines, making clear what features the authors refer to and in which figure panels.

The authors also suddenly start talking about the geographical location of the 2 floats, without any preamble or reference to Figure 1. I suggest to remind the reader of the location and trajectory of the 2 floats before discussing chl features detected by each.

8.  Line 244: What is meant by "compensational irradiance"? I suggest to clarify in the text.

**Discussion**

9.  Figure 8: I don't think it's that useful to have the NO3 depicted in both panels of the figure if the profile is exactly the same.

**Technical Comments**

**Introduction**

1.  Line 35: "The biomodelling study by Kubryakova et al. (2018)" → I would not use the word "biomodelling," this is definitely not a standard term that is recognised by the community. Biogeochemical or ecosystem model would be more appropriate (or just "modelling").

2.  Line 45: "nutrients" should be nutrient.

3.  Line 54: change ", documented for the Black Sea in … " to " which has been documented in the Black Sea (*references)* "

4.  Throughout the manuscript please change "buoys" to floats. The use of buoys may lead to confusion since BGC-ARGO are floats.

**Methods and Data**

5. Figure 1: I suggest to only show the isobaths that are labelled (2000, 1600, 1000, 200 m), since as the figure is now there are so many that it becomes meaningless.

6. Line 125: What is the depth of the reference density used for the MLD calculation?

**Results**

7. Figure 4: Which float is the data taken from? If it is an interpolation of both then the method of interpolation must be provided. Add details to the caption.

8. Figure 5: State in the caption how the difference is computed, is it 2016 - 2017 or the other way around? Following this, it would also be helpful to say what positive and negative values mean, e.g. "positive values indicate the chl values are higher in 2017"

9. Figure 7:  It is unclear what is being compared here. Are the red lines 2016 and blue 2017? Or do they represent different floats? Please clarify in the caption, and also add legends to the figures.

10. Line 154: conventional should be *convectional.*

11. Line 213: "Ten-daily diagram… " Change to "Fig. 6a-d shows the same features at a higher frequency of 10 days..."

12. Line 233: "Jule-September"

**Discussion**

13. Lines 291 - 292: " *Entrained in winter period nutrients and the rise of the irradiance causes the following spring growth of phytoplankton.* " Reword as: "Winter entrainment of nutrients, followed by increased irradiance in spring, is known to lead to enhanced phytoplankton growth."

---

## Author Comment (AC1) · 12 Dec 2020

Nicolas Mayot, we would like to thank You for comments and valuable and constructive suggestions for improving the paper.

Please note that this response to the review is accompanied by corrected figures (Fig. 2-4, 6, 8). These figures are given here according to the numbering in the revised manuscript. Figures R1-R4 will not be inserted in the article. They are provided here to explain the answers.

General comments (GC)

GC1: "Authors wrote several times in the manuscript that in the Black Sea, as in other oligotrophic basins, the nitracline is closely connected to an isopycnal (for examples, see lines 80, 156, 161 and 287). When looking at figure 2 c-d, we can clearly see that isopynals between 1014 kg/m3 and 1014.4 kg/m3 (associated with the nitracline, see line 158 and figure S1) are shallower in summer 2017 than in summer 2016. Therefore, in summer 2017 the nitracline (or nutricline) is shallower than in summer 2016. The low chlorophyll-a concentration in 2017 between 40-60 m is probably due to a change in in the depth of isopycnals and not to a "rise of light attenuation", as argued by authors (line 387).

Answer GC1. In fact, the concentrations of nitrates and phosphates rise gradually with depth reaching their subsurface maximum at 1015.5 kg/m3 (see Fig. R1 below). Nutricline defines the area of the highest gradients of nutrients concentration but not the position of their maximum. Therefore, at deeper isopycnal layers (1014.6-1015.5 kg/m3), the concentration of nutrients is even larger than at 1014-1014.4 kg/m3. According to Your assumption, the Chl should have been higher in the deeper layers in 2017, when the isopycnals were raised. However, according to the Bio-Argo measurements, Chl was lower in deep layers in 2017 (see Fig. 3-5 in the manuscript).

In the manuscript, it was written: "For example, the concentration of nitrates begins to gradually increase below the isopycnal of 1014 kg/m3 (see Supplementary Fig. S1), and rises more sharply below the isopycnal of 1014.4 kg/m3 where the upper part of the nutricline is located (Konovalov & Murray, 2001)". To avoid misunderstanding, we have rewritten this phrase as "The concentration of nitrates begins to gradually increase below the isopycnal of 1014 kg/m3, and reach its maximum at $\sigma\sim$=15.5 kg/m3 (Konovalov & Murray, 2001)".

To answer Your question, we have added to the paper the variability of the Chl in density coordinates (see revised Fig. 4c, d below). As it is seen, there is no correlation

between Chl and density. For example, in March-May at $\sigma$=1014.5 kg/m3, Chl in 2017 was higher than in 2016, while at the summer period Chl in 2016 was higher than in 2017. This indicates that Chl variability is not directly related to isopycnals' position, which is related to the crucial role of light penetration, which depends on the bio-optical properties of the water in upper layers and defines the isopycnal layers, where phytoplankton can grow.

We have revised the Fig. 4 and its description to the paper.

GC2: Line 343 – "The closeness of DCM and nutricline increases the nutrient fluxes in the summer of the years with warm winter, which compensates for their decrease in winter caused by weak convective entrainment." Such a statement is false because the authors did not discuss differences in the depth of the nutricline between summer 2016 and summer 2017.

Answer GC2. We agree that this statement was inaccurate. So we have rewritten it as "The closeness of DCM and deep isopycnal layers with high amount of nutrients increases the nutrient fluxes in the summer of the years with warm winter, which compensates for their decrease in winter caused by weak convective entrainment."

To demonstrate this fact, we have added the diagrams of the seasonal variations of Chl in density coordinates in 2016 and 2017 (Fig. 4c, d) to the paper. As You can see, in the warm period of 2016, higher values of Chl was detected in deeper isopycnal layers (reaching 1014.7 kg/m3), which contain higher concentrations of nutrients. That is why we conclude that the DCM was closer to the nutricline in the summer of warm 2016.

We have added this information to the text: "The same features are seen in the Fig. 4c, d, which demonstrates the variability of Chl in $\sigma$-coordinates, where $\sigma$ is potential density. Chl in 2017 was higher in upper isopycnal layers. However, in 2016 relatively large values of Chl were observed in the deepest isopycnal layers. Particularly in 2016, in summer months, Chl at $\sigma$=1014.5-1014.7 kg/m3 exceeded 0.2 mg/m3, whereas, in 2017, it was near zero. These values of density 1014.5-1014.7 kg/m3 were higher than

that density of waters entrained to the surface in winter (revised Fig. 2c below), which was less than 1014.2 kg/m3".

"Particularly, in the considered case in warm 2016, large values of PAR (more than 3 $\mu$mol photons m-2 s-1) reached 40-60 m depth (Fig. 4b), corresponding to $\sigma$=1014.5-1015 kg/m3 (Fig. 4d). In these isopycnal layers, the concentration of nitrates sharply increases (Konovalov& Murray, 2001; TuÄ§rul et al., 2015). These values were significantly higher than that density of waters entrained to the surface in winter (Fig. 2c), which was less than 1014.2 kg/m3. Therefore, Fig. 4d evidence that in the summer of 2016, the penetration of light induces the rise of Chl directly in the nutricline waters, which were not affected by winter convective mixing."

We would like to thank You for this comment, because it helps us to demonstrate our results more clearly.

GC3: In addition, when looking at figures 4a-b, chlorophyll-a concentration at the DCM is higher in 2017 than in 2016. Considering that isopycnals were shallower in 2017 than in 2016, and based on figures 4e-f, one could argue that the light level at the nutricline is higher in 2017 than in 2016. A higher light level at the nutricline could induce a higher phytoplankton production and biomass in the DCM, as observed in 2017. Such mechanism, clearly explained in the comprehensive review of Cullen (2015), is crucial and need to be discussed".

Answer GC3. We agree that Chl at DCM was higher in 2017 and 2016. However, in 2016, DCM was wider and occupied a larger layer, as in depth-coordinates (Fig. 4a, b, 5 (in the manuscript)), as in density-coordinates (revised Fig. 4c, d).

The rise of the phytoplankton depends on the absolute values of nutrients concentration, not on their gradients (defining nutricline). It is important to note that the maximum of nutrient concentrations in the Black Sea and other ocean areas is located far below the euphotic layer (see answer on GC1, GC2). Therefore, when light penetrates to deeper isopycnal layers (as in 2016, see revised Fig. 4c, d), Chl values should be

larger. That is why, as we show in revised Fig. 3b, in fact, the integral concentration of Chl was higher in the summer of 2016 than in the summer of 2017.

GC4: "When studying DCM, it is important to know if the DCM is also a subsurface phytoplankton biomass maximum. Because, changes in chlorophyll-a concentration at a DCM could be induced by variations in intracellular chlorophyll-a concentration or by changes in phytoplankton biomass. In the current version of the manuscript, this point is not discussed".

Answer GC4. We agree and have extended the Discussion on this topic: "It should also be mentioned that the vertical distribution of phytoplankton biomass often does not coincide with the distribution of Chl (Finenko et al., 2002, 2005; Krivenko, 2010; Churilova et al., 2017). Particularly, in the summer period, the peak of the phytoplankton biomass is usually located above the maximum of Chla (Finenko et al., 2005; Krivenko, 2010; Stelmakh & Babich, 2006). At this time, large diatoms such as Pseudosolenia calcaravis and Proboscia alata with high biomass and low cellular Chl prevailed in phytoplankton layers (Mikaelyan et al., 2018; Silkin et al., 2019). At the same time, the deepest phytoplankton in summer mainly consists of small flagellates and unicellular cyanobacteria (Rat'kova, 1989; Churilova et al., 2019; Mikaelyan et al., 2020), which have very high specific cellular pigment content. As a result, the biomass may be low, whereas the Chl is rather high. The phytoplankton after cold and warm winter grows at different depths, and, therefore, the environmental (light and nutrients) conditions change significantly. Deepening of the euphotic layer may promote the growth of species adapted to low light with low biomass and high Chl content (Falkowski & La Roshe, 1991; MacIntyre et al., 2002; Latasa et al., 2017), which may have an advantage in the years with warm winters".

GC5: "When using data from biogeochemical-Argo floats equipped with an ECO Triplet (as here, line 110), measurements of particulate backscattering coefficient (bbp), a proxy of the particulate organic carbon, are available. These measurements have been used in a paper published by one of the authors (Kubryakov et al., 2019). The

manuscript will be more comprehensive if those measurements could be added to the current analysis".

Answer GC5. Thank You for this advice. We agree and have added the diagram of the bbp difference in the revised version of the manuscript (new Fig. 6d). This figure complemented our results. It clearly demonstrated the rise of bbp in the summer period in the upper layer, related to the very intense coccolithophores bloom in 2017. We have added this information to the text: "This fact reflected in the strong rise of the bbp in 2017 (see new Fig. 6d), which in summer period of 2017 reached its maximum over 2014-2019 period (Kubryakov, Mikaelyan, et al., 2019). The rise was observed in the upper layers (0-30 m), which are usually occupied by coccolithophores bloom (Mikaelyan et al., 2005; Kubryakov, Mikaelyan, et al., 2019). In May-August, bbp in the upper 0-20 m layer was higher on 0.015 m-1 (new Fig. 6d) or threefold higher compared to 2016. At the same time, new Fig. 6d shows the decrease of bbp in the deeper layers (30-60 m) in the summer of 2017. Coccolithophore blooms usually are not observed in these deep layers, and this decrease can be attributed to the decrease of phytoplankton biomass at these depths in 2017, which is in agreement with Chl variability (new Fig. 6a)".

Minor comments (MC)

MC1: "Line 11 – "caused an increase of Chl in winter up to 0.6-0.7 mg/m3 compared to a warm winter of 2016": It is unclear. There is an increase of Chl in winter in both years".

Answer MC1. We have rewritten this part of the abstract: "In cold 2017, nutrient-rich waters from deeper isopycnal layers were entrained to the surface. As a result, Chl during early-spring bloom in 2017 was in 1.5-2 times higher (0.6-0.7 mg/m3) than in warm 2016 (0.4-0.5 mg/m3)".

MC2: "Line 19 – "more productive." That is a suggestion. Chlorophyll-a concentration measurements are not primary production estimates".

Answer MC2. We agree and have corrected it as "in the year with intense winter mixing, Chl in upper layers is higher."

MC3: "Line 24 – "With the rise of stratification and irradiance vertically. . ." A comma is needed after the word "irradiance."

Answer MC3. Corrected.

MC4: "Line 26 – "After the bloom, part of the nutrients. . .fuel the phytoplankton bloom". How many blooms are there in a year?".

Answer MC4. Corrected on "which fuels the phytoplankton growth in the warm period of the year."

MC5: "Line 27 – "nitrocline." It is a wrong spelling, used several times in the manuscript. Need to be replaced everywhere by "nitracline" or "nutricline." However, here it could be replaced by "euphotic zone."

Answer MC5. Thank you. In the revised manuscript, "Nitrocline" was replaced everywhere by "nutricline." At line 27, it was replaced by a "euphotic zone."

MC6: "Line 31 – ". . .convection in the Black Sea, impact on the. . .". Need to be replaced by ". . . convection in the Black Sea impacts on the. . .".

Answer MC6. Thank you. It was replaced.

MC7: "Line 35 – replace "biomodelling" by "modelling".

Answer MC7. It was replaced.

MC8: "Line 45 – The diapycnal diffusivity could be an important physical process for nutrient supply. Maybe it is the most important in summer".

Answer MC8. We have corrected the phrase as "The nutrients entrainment in the euphotic layer in the warm period is strongly related to the diapycnal mixing (or diffusivity) enhanced by short-period mixing events (Williams & Follows, 2003), such as storms

(Iverson et al., 1974; Zhang et al., 2014; Kubryakov, Zatsepin, et al., 2019), wind-driven or dynamic upwelling (McGillicuddy et al., 1998; Mikaelyan et al., 2020)".

MC9: "Line 52 – The relationship between expressions: "high values of irradiance may cause photoinhibition", "photoadaptation" and "increase of Chl content per cell" is unclear".

Answer MC9. Thank You. In the revised manuscript, this phrase was corrected as "At the same time, high values of irradiance cause the decrease of Chl and fluorescence near the surface (Platt et al., 1982) due to several effects including non-photochemical quenching, photoinhibition, and photoacclimation (Falkowski & Raven, 2013). The later lead to the decrease of Chl content per cell (MacIntyre et al., 2002), which is observed in the upper layers of the Black Sea in the summer period (Finenko et al., 2002, 2005; Silkin et al., 2013; Churilova et al., 2019)".

We also have extended the Discussion of the photoacclimation in the Discussion part of the revised manuscript (please see answer GC4).

MC10: "Line 66 – Replace "Bio-Argo buoys" by "biogeochemical-Argo floats". Throughout the manuscript, the word "buoys" should be replaced by "floats".

Answer MC10. Line 66 is corrected. Throughout the manuscript, the word "buoys" was replaced by "floats."

MC11: "Line 89 – "Long-term rise of temperature observed globally (Behrenfeld et al., 2016)" This study did not focus on the long-term rise of temperature."

Answer MC11. Thank You. We agree and have changed the reference on the proper one: Boyce, D. G., Lewis, M. R., & Worm, B. (2010). Global phytoplankton decline over the past century. Nature, 466(7306), 591-596.

MC12: "Line 121 – "The letter z is missing in the parentheses".

Answer MC12. We have corrected this formula and excluded $\lambda$ from it, as we use here

only Kd of PAR: $K_d(z) = \ln(a((E_d(z+dz))/(E_d(z)))/dz$

MC13: "Line 124 – All measurements from Argo floats have quality control flag values. Did authors check them?"

Answer MC13. Yes, we checked QC flags. However, our analysis showed that many data marked as bad (QC=4) was, indeed, reasonable. That is why we carefully checked all the bio-optical data using visual analysis, as it is shown in Fig. R2 below. In fact, there were almost no spikes in the data, with the exception of deep layers with depths more than 100-200 m, which are out of the zone of interest for our study.

MC14: "Line 125 – Information about the process to obtain potential density values from in situ temperature and salinity measurements are missing".

Answer MC14. Potential density was computed from data of temperature and salinity on the base of UNESCO formulae using seawater Matlab codes (Morgan, 1994). We added this information to the text.

MC15: "Figure 2 – Is it in situ or potential temperature? Potential density? The colorscale or isolines are inconsistent. For example, in panel C, the isopycnal 1014.2 has a yellow background and in the panel D has an orange background.

Answer MC15. This is in-situ temperature and potential density. This information is added to the revised Fig. 2 caption (see it in answer GC2). We also redraw Fig. 2c, d to show the proper position of the contour lines. Thank You for this mark.

MC16. Here and throughout the manuscript, could the authors use another colormap, not a rainbow one: "In the 2007 IEEE article, Rainbow Color Map (Still) Considered Harmful, authors David Borland and Russell M. Taylor II from the University of North Carolina at Chapel Hill stated, "The rainbow color map confuses viewers through its lack of perceptual ordering, obscures data through its uncontrolled luminance variation, and actively misleads interpretation through the introduction of non-data-dependent gradients."

Answer MC16. Thank you for the information. Until now, we have not paid attention to this. We have read the article which You cite. In conclusion, the authors wrote: "The selection of the best color map depends so critically on the data set and addressed questions that there is not a single best choice, but rather a collection of sets with different characteristics. The best solution would present the user with a choice whenever a color map is created, listing best types for each circumstance".

Interested in this question, we have read the results of one more recent study (Khairi Reda, Pratik Nalawade, and Kate Ansah-Koi. 2018. Graphical Perception of Continuous Quantitative Maps: the Effects of Spatial Frequency and Colormap Design. Proceedings of the 2018 CHI Conference on Human Factors in Computing Systems. Association for Computing Machinery, New York, NY, USA, Paper 272, 1–12. DOI:https://doi.org/10.1145/3173574.3173846). The authors concluded that "...data reveals that, counterintuitively, rainbow is robust for estimating a smoothly varying quantitative attribute, regardless of spatial complexity. Moreover, rainbow provides good support for gradient estimation at high spatial frequency. These two tasks correspond to elementary visual analytic primitives, including characterizing distributions, determining ranges, and filtering [1]. Moreover, studies show that when experts attempt to form a mental model about a visualization, they first go through a time-consuming process of extracting quantitative data "at a rather detailed level" [38].

For instance, a weather forecaster will lookup pressure and wind changes, estimating current readings at landmark locations in the map before making a forecast. Our data suggests that rainbow provides good support for these tasks, making it a potentially reasonable choice for weather forecasters...".

In our manuscript, we use rainbow to estimate a smoothly varying quantitative attribute (e.g. temperature, density etc.), so we would like to keep this colormap. In the future, we will take the information you provided into account to choose the colormap.

MC17: "Line 154 – "conventional mixing", is it the right term?".

Answer MC17. Thank You. It was changed to "convective mixing."

MC18: "Section 3.1, Figure 2 and Figure 3 – Regarding physical parameters, there are no information or comments about potential differences between the two floats for a same year due to differences in their geographical locations. Are there any differences in the vertical distribution of temperature, isopycnals and MLD between the two floats for a same year? In addition, time series of temperature and density for each float should be added in supplementary materials. The sentence, line 101: "As we will show below, despite these differences in the geographical position both buoys show similar results concerning the paper topic" needs to be supported by figures and values for all physical and bio-optical parameters used here".

Answer MC18. Yes, there were differences in the physical parameters measured by these two floats. This was related to their different geographical position. Float #7900591 was moving mostly in the center of the sea, corresponding to the center of cyclonic gyres. Therefore, isopycnal positions were located closer to the surface according to its measurements compared to floats #6901866, located in the downwelling area (Fig. R3). The same is observed for the temperature variability (Fig. R4). The cold intermediate layer (white color in Fig. R4) is situated deeper over the continental slope (float #6901866) than in the basin's central part (float #7900591).

Despite these differences in physical parameters, Fig. 4 evidences that the main feature of the interannual variability in Chl distribution is similar for both floats: in the cold 2017, Chl was higher in the upper layers, while in warm 2016, it was higher in deeper layers.

For clarity, in the revised manuscript, we have rephrased this sentence: "As we will show below, despite these differences in geographical locations, both floats show similar results concerning year-to-year variability of Chl vertical distribution."

This is also clearly demonstrated below in Fig. S2, which shows average annual Chl profiles measured by two floats.

We decided not to include the Fig. R3 and R4 in the manuscript as they are not crucial for the goal of the paper but may increase and overload the manuscript.

MC19: "Line 201 – "productivity" chlorophyll-a concentration is a proxy for phytoplankton biomass not phytoplankton production".

Answer MC19. It was changed on "to an increase of Chl."

MC20: "Line 256 to 259 – This is a suggestion, it should be discussed in the discussion section".

Answer MC20. We agree and have moved this paragraph to the Discussion.

MC21: "Figure 7 – Why yearly-averaged profiles and not only summer (and winter) profiles".

Answer MC21. Thank You for this advice. We have inserted summer (and winter) profiles to the text (see revised Fig. 7, now it is Fig.8). This helps to visualize our results more clearly.

MC22: "Line 267 and figure 3b – It would be better to describe time series of depth integrated chlorophyll-a concentration and not column-averaged".

Answer MC22. We have added the graph of the time variability of depth-integrated Chl to Fig. 3 and added their values in the text.

MC23: "Line 274 to 276 – This is a suggestion, it should be discussed in the discussion section".

Answer MC23. We agree and excluded this phrase from the results.

MC24: "Line 308 – "the euphotic layer deepens, and the surface layer becomes overilluminated" It would be better to use the euphotic depth definition. What is the meaning of "over-illuminated"?".

Answer MC24. We have rewritten this phrase as: "In the warm period of a year

PAR strongly increases, light penetrate to deeper layers. At the same time, PAR in the surface layers becomes too strong, which leads to the photoacclimation of phytoplankton and a decrease of Chl in upper layers (Falkowski & Raven, 2013)".

Please also note the supplement to this comment:
https://bg.copernicus.org/preprints/bg-2020-366/bg-2020-366-AC1-supplement.pdf

[Figure]

**Fig. 1.** Fig. R1. Multiannual-averaged profile of nitrates (blue) and phosphates (red) in the central part of the Black Sea in $\sigma$-coordinates

[Figure]

**Fig. 2.** Fig. R2. Example of Chl profile for 31 August 2020 with QC flags shown by colors.

**Fig. 3.** Fig. R3. Time variability of density according to measurements of the float #7900591(top) and float #6901866 (bottom)

[Figure]

**Fig. 4.** Fig. R4. Time variability of temperature according to measurements of the float #7900591(top) and float #6901866 (bottom)

[Figure]

**Fig. 5.** (Revised) Figure 2: Monthly vertical diagram of temperature in 2016 (a) and 2017 (b), potential density in 2016 (c) and 2017 (d).

[Figure]

**Fig. 6.** (Revised) Figure 3. Full caption is in the pdf-version of the response

[Figure]

**Fig. 7.** Fig. 4: Seasonal variability of Chl in 2016 (a,c) and 2017 (b,d) in z-coordinates and density coordinates. The black line-the border of the photic zone (Ed=3 $\mu$mol photons m-2 s-1), the red line-MLD

[Figure]

**Fig. 8.** Fig. 6: Seasonal diagram of a difference of vertical distribution of Chl (a), Kd (b), bbp (c) between 2017 and 2016. Difference in PAR ($\mu$mol photons m-2 s-1) distribution between 2016 and 2017 (d)

[Figure]

**Fig. 9.** Figure 8: Average profile of Chl in 2016 (red line) and 2017 (blue line) in winter (January-March) (a), summer (June-August) (b) and annually-averaged (c).

[Figure]

**Fig. 10.** Supplementary Figure S2: Average profile of Chl in 2016 and 2017 by the measurements of the float #7900591 (a) and float #6901866 (b).

---

## Author Comment (AC3) · 12 Dec 2020

We would like to thank Vladimir Silkin for comments and valuable and constructive suggestions for improving the paper.

Comments and Recommendations (CR)

CR1. "In the Introduction, it is necessary to formulate the research objectives more clearly. In particular, the sentence on line 68 should be moved to the end of the Introduction".

[Figure]

Answer CR1. Thank You for this comment. We agree and have rewritten the Intro-duction part to formulate the research objectives more clearly. We also moved part of the text from Introduction to the "Section 2.1 General information about the study area". The revised part of the Introduction: "In our study, we use the measurements of biogeochemical-Argo floats in the Black Sea to investigate the effect of winter mixing and irradiance on the vertical distribution of Chl in two years with contrasting winter conditions – warm in 2016 and cold in 2017. The analysis revealed that the decrease of nutrient fluxes in the years with warm winter is partly compensated in the summer period, when, in conditions of low Chl, light is able to penetrate in the deeper layer with a larger amount of nutrients. The structure of the manuscript is as follows. Section 2 gives information about the study area and used data and methods. Section 3 presents the differences between main physical (Section 3.1) and bio-optical (Section 3.2) char-acteristics in the warm 2016 and cold 2017 years. Discussion and conclusions are presented in Sections 4 and 5."

CR2. "On lines 78 - please provide a reference".

Answer CR2. We have added the reference in this part of the text – (Titov, 2004).

CR3. "In the Results, provide only the authors' results and move the comments and reference to the Discussion (Lines, 144-145, 157-158, 160-163, 173-183, 201-205, 2016, 234, 244, 265)".

Answer CR3. Thank You. We agree and have moved part of this text to the Discussion and part to the "Section 2.1 General information about the study area". We kept the minimum amount of the references in the Results Section, only those needed to explain the obtained features of the Black Sea bio-optical properties.

Individual comments and recommendations (ICR)

ICR1. "Line 19 and 201-chlorophyll concentration and productivity are not the same things".

Answer ICR1. We agree and have corrected it as: "in the year with intense winter mixing, Chl in upper layers is higher" on line 19 and changed on "to an increase of Chl" on line 201.

ICR2. Line 35 - "the biomodelling study" rewrite as "the modeling study".

Answer ICR2. It was rewritten.

ICR3. "Line 85 - Cold intermediate layer mark as CIL and use it later in the article".

Answer ICR3. The cold intermediate layer was marked as CIL on line 85. And since now CIL is used in the paper.

ICR4. "Lines 50-55. Rewrite these sentences "At the same time, high values of irra-diance may cause photoinhibition and decrease of Chl near the surface (Platt et al., 1982) caused by several effects including non-photochemical quenching, photoinhibi-tion, and photoadaptation (Falkowski & Raven, 2013). The latter is partly associated with the increase of Chl content per cell (MacIntyre et al., 2002), documented for the Black Sea in (Finenko et al., 2002, 2005; Churilova et al., 2019)". Here it would be best if you replaced photoadaptation with photoacclimation, since changes in the chlorophyll content in the cell are acclimation. Further, the chlorophyll content in the cell increases when the light intensity decreases. Therefore, it decreases at the surface".

Answer ICR4. Thank You. We agree and have rewritten this sentence. In the revised manuscript: "At the same time, high values of irradiance cause the decrease of Chl and fluorescence near the surface (Platt et al., 1982) due to several effects including non-photochemical quenching, photoinhibition, and photoacclimation (Falkowski & Raven, 2013). The later lead to the decrease of Chl content per cell (MacIntyre et al., 2002), which is observed in the upper layers of the Black Sea in the summer period (Finenko et al., 2002, 2005; Silkin et al., 2013; Churilova et al., 2019)."

ICR5. "Line 110-111. Rewrite this sentence".

Answer ICR5. We have rewritten this sentence as "We used the product

"CHL_ADJUSTED", which includes the correction on non-photochemical quenching (Roesler et al., 2017) and the correction on the contribution of fluorescence by non-algal organic matter (Xing et al., 2017)". The data was downloaded from IFREMER data archive ( ftp://ftp.ifremer.fr/ifremer/argo).

ICR6. "Line 121. More correct to write Kd(z) rather than Kd($\lambda$) since this parameter changes with depth".

Answer ICR6. We have corrected this formula and excluded $\lambda$ from it, as we use here only Kd of PAR K_d (z)=lnâĄą((E_d ( z+dz))/(E_d ( z) ))/dz

ICR7. "Line 129-decode DAC".

Answer ICR7. We have rewritten this sentence as: "Bio-Argo data have a high time resolution (from 1 to 5 days) and vertical resolution (1 m), is regular and is publicly available at http://doi.org/10.17882/42182 or can be downloaded from the Data Assembly Centers (such as http://www.coriolis.eu.org/)."

ICR8. "Line 187-there is no dot at the end of the sentence".

Answer IC8. Thank you, the dot was added.

ICR9. "Line 315-chlorophyll and biomass are not the same things".

Answer ICR9. We have rewritten this sentence as "On the opposite, in the year with relatively low winter nutrient fluxes (as in 2016 in the Black Sea), the growth of phytoplankton and related to it Chl decreases".

ICR10. "Figure 4 is missing the dimension for PAR".

Answer ICR10. We have added the units to all figure captions.

ICR11. "In the caption to Figure 7, there is no designation of the curves (red and blue)".

Answer ICR11. Thank you. The curves designation was added: red line – 2016, blue line – 2017.

Please also note the supplement to this comment:
https://bg.copernicus.org/preprints/bg-2020-366/bg-2020-366-AC3-supplement.pdf

---

## Short Comment (SC3) · 13 Dec 2020

All my comments were taken into account, and the authors made appropriate corrections to the text. I believe the article meets the requirements of the journal.

---

## Author Comment (AC4) · 29 Dec 2020

Elena A. Kubryakova and Arseny A. Kubryakov

elena_kubryakova@mail.ru

Response to reviewer #2

Reviewer: "The authors investigate the drivers of differences in the vertical distribution of chlorophyll-a between 2016 and 2017 in the Black Sea using BGC-ARGO data. A key feature of interest in the vertical distribution is the so-called deep chlorophyll maximum (DCM), which the authors show is deeper and less intense in 2016 than in 2017. They account for this difference by arguing that cold atmospheric conditions in

the winter of 2017 led to convective mixing and nutrient entrainment, thus increasing winter production. It is then argued that this increased production led to enhanced self shading in 2017, which accounts for why the DCM is shallower compared to 2016. In general, I agree with other reviewers that the hypothesis presented is interesting and could represent a significant contribution to the question of what factors control the DCM. However, I also agree that currently the authors do not present sufficient evidence to support their hypothesis. Furthermore, the methodology requires some important revisions which I explain below. I therefore recommend that the following revisions be undertaken prior to publication".

Authors: First, we would like to thank the Reviewer for comments and constructive suggestions for improving the paper.

General comments (GC).

GC1. "All monthly averaging should be removed or only added to supplement the higher frequency data. This is actually why there is little difference seen in the MLD between the 2 years - the differences have been averaged out. Below I show an example of temperature profiles for early February comparing the 2 years. Here it is clear that the MLD is deeper in 2017 by ∼20 m, although if you average over the whole month you won't see much difference. This highlights that the phenomenon being investigated occurs at much higher frequency than monthly, which needs to be taken into account in more detail than is currently done".

Answer GC1 Unfortunately, we can not fully agree with the suggestion that monthly-averaged data cannot be used in the study. We agree that the short-period oscillations of Chl and the reasons for their variability is a very important task. The detailed investigation of year-to-year seasonal changes of Chl in the Black Sea in 2014-2019 was made in our recent study (Kubryakov, A. A., Mikaelyan, A. S., Stanichny, S. V., Kubryakova, E. A.: Seasonal Stages of Chlorophyll‐a Vertical Distribution and Its Relation to the Light Conditions in the Black Sea from Bio‐Argo

Measurements, Journal of Geophysical Research: Oceans, 125, e2020JC016790, https://doi.org/10.1029/2020JC016790, 2020). We hope that we will be able to investigate the reason for even more high-frequency variability of Chl in our future studies. Particularly, one such study related to the impact of intense storm on the anomalous rise of Chl in August 2015 on the base of Bio-Argo data was carried out in (Kubryakov, A. A., Zatsepin, A. G., and Stanichny, S. V.: Anomalous summer-autumn phytoplankton bloom in 2015 in the Black Sea caused by several strong wind events, Journal of Marine Systems, 194, 11-24, https://doi.org/10.1016/j.jmarsys.2019.02.004, 2019).

However, in the present manuscript, we investigate the reasons, which can explain why DCM in one year was deeper than in another year. That is why we are focusing on annual time scales and need to average the data. The time-averaging is a typical oceanographic technique that is widely used for the investigation of processes on different time scales (see, for example, Fig. 8 and 10 in (Mignot, A., Claustre, H., Uitz, J., Poteau, A., D'Ortenzio, F., Xing, X. (2014). Understanding the seasonal dynamics of phytoplankton biomass and the deep chlorophyll maximum in oligotrophic environments: A Bio-Argo float investigation. Global Biogeochemical Cycles 28: 856-876 | DOI: 10.1002/2013gb004781).

Below, we show the examples explaining this statement, similar to the one presented by the Reviewer. In Fig. R1 You can see the data of only one float (#690186) for the February month of 2016 and 2017. In the left figure, we chose profiles, where MLD in 2016 was larger than in 2017. This figure can lead to a conclusion that MLD was higher in 2016. In the second figure, we chose profiles, where MLD in 2016 was larger than in 2017. This figure can lead to a conclusion that MLD was higher in 2017. In 2017 larger amount of profiles have higher MLD, by there were also opposite cases. This figure presents only the measurements of one float. Therefore, to understand in what years MLD was deeper, we need to average the data.

At the same time, we agree that it may be helpful for the paper to give information about maximal MLD, density, and minimum temperature observed in both years. These
values more clearly define the maximum intensity of winter convection, detected by float measurements. We have added this information to the text:

lines 172-173: "Minimal temperature at 5 m depth detected by Bio-Argo floats was equal to 7.8°C in 2016 and 5.5°C in 2017."

lines 189-190: "Maximum density at 5 m depth detected by Bio-Argo floats was equal to 1014.44 kg/m3 in 2016 and 1014.70 kg/m3 in 2017."

line 202: "Maximum mixed layer depth reached 65 m in 2016 and 85 m in 2017."

GC2. "Similarly to point 1 above, the data should be presented with as little interpolation as possible. It is clear from figures 2, 4, 5 and 6 that some kind of spatiotemporal interpolation has been done to produce such highly "smoothed" plots. Below I show an example of how the chl-a data look for float 6901866 with a minimal amount of interpolation (here I only use a linear interpolation in the "depth" dimension for the missing data, and gaps of greater than 5 m are not interpolated) I suggest to change the figures to something more like this, which portrays the data more accurately.

Answer GC2. We do not use any spatiotemporal interpolation in Fig. 2, 4, 5, and 6. The Fig. 2, 4, 5 present monthly-averaged data. As it is stated in Section 2.2, we only interpolated data vertically on a 1 m grid (similar as You do). In Fig. 7, we use ten-daily averaging to obtain data on the regular time grid. This is stated in the revised text in line 262.

However, we use a different visualization technique. If it is a Matlab, we prefer to use contour plot, and we think You use "imagesc" (or "pcolor"). We try to reproduce Your code approximately and have below attached the figure of Chl variability (5-days binarization) for float #6901866 plotted with a use of "imagesc" (Fig. R2b). As You can see, Fig. R2a and R2b are very similar. For the comparison, Fig. R2c presents the same data using contour function, other colorbar, and color limits. The same data looks different when using different visualization techniques.

Both functions, "contour" and "imagesc" ("pcolor"), are widely used. Both of them use interpolation. "Imagesc" ("pcolor") use the nearest interpolation, while "contourf" use continuous one. Discrete and continuous colorbar also, of course, play their roles.

We send You the code in Matlab below. Please check if You will have the same result.

Figure

contourf(d,-z, chl,100,'lines','none')

datetick('x')

caxis([0 1])

GC3. Here it is clear that the high chl-a values seen in winter of 2017 are actually composed of 2 short periods (10-15 days) of elevated growth, one in December and another stronger one in March. Figure 2 in the current manuscript makes it seem like one long period of sustained growth. Figure 6 does actually show these 2 pulses, but since 2016 and 2017 are split into separate panels one cannot easily see the 2 distinct growth periods. The plot above also shows that the DCM is most intense (highest chl-a) in the autumn of 2018 - it might be interesting to look into why this is the case".

Answer GC3. We agree with this comment. Yes, there are two peaks in November 2016 – March 2017. Such two peaks are the usual pattern of the seasonal Chl dynamics in the Black Sea. Actually, three peaks of Chl are detected in the Black Sea throughout the year: February-March peak, summer peak, and late autumn-early winter peak in November-January. Both February-March and November-January peaks are related to the intensity of winter convection. They are separated by the minimum in February, which is related to the deepening of the mixed layer below the euphotic layer (Sverdrup, 1954). These features of the seasonal changes in Chl were in detail investigated in our recent study (Kubryakov et al., 2020). In (Kubryakov et al., 2020), we also demonstrate and discuss the year-to-year difference in Chl variability in the 2014-2019 period.

We agree that both February-March and November-January peaks were strong in the

cold winter of 2016-2017. We have added this information in the revised version of the manuscript: "In addition, float #6901866 detect significantly more intense late autumn Chl bloom in November-December of 2016. This seasonal bloom is also driven by the winter mixing (see Finenko et al., 2014; Mikaelyan et al., 2017; Kubryakov et al., 2020), which was more intense in the cold season of 2016-2017." (lines 269-272).

Kubryakov, A. A., Mikaelyan, A. S., Stanichny, S. V., & Kubryakova, E. A. (2020). Seasonal Stages of Chlorophyll Řa Vertical Distribution and Its Relation to the Light Conditions in the Black Sea from Bio ŘArgo Measurements. Journal of Geophysical Research: Oceans, 125, e2020JC016790. https://doi.org/10.1029/2020JC016790

GC4. "I follow the argument that the upliftment of isopycnals is associated with a rise in the nutricline and therefore nutrient entrainment into the MLD. However, I would argue that simply referring to other literature where this relationship has been established is not sufficient to say that it has occurred in the present case. Since this entrainment of nutrients is key to the argument being made, it follows that it should be explicitly shown with data. Here I recognise that the nitrate data may be biased in these particular floats as the authors have suggested. However, the important point is that nitrate concentrations should be higher in the cold 2017 year, so biases in the concentration may not preclude the use of this data (since we look for relative differences, not absolute values). So long as the bias is properly taken into account I would argue that the data should be used to support the argument. If the data are really not appropriate, perhaps other proxies for entrainment of deep water could be used (e.g. dissolved oxygen)?"

Answer GC4. We made such a comparison to answer Your comment. The graph below shows the seasonal variability of nitrates in surface layers in different years (Fig. S1). First, we notice that it approve a higher nutrient amount in winter of 2017 than in 2016. However, we think that we can not refer to this data, as it shows completely incorrect values of NO3. Bio-Argo derived values of NO3 was in 10 times higher than the data from numerous in-situ studies (see Fig. S1a, b).

[Figure]

Supplementary Fig. S1: (a) the multi-annual average vertical profiles of nitrate (NO3) and phosphate (PO4) in $\sigma$-coordinates for October, the month preceding the onset of intense winter convection from in-situ MHI data archive; (b) seasonal variability of NO3 at 1 m depth in 2015-2020 according to Bio-Argo measurements.

This is the most significant problem, which does not allow to publish such data. There are some other problems: incorrect seasonal variability with maximum in summer and minimum in winter (Fig. S1b); long-term trend of NO3 (Fig. R3), which indicate the possible drift of the sensor.

Personally, our analysis of Bio-Argo optical NO3 measurements indicated that they were able to "feel" the lower boundary of nutricline, but not the proper values in upper layers. We understand that this method is experimental and hope the Bio-Argo team will be able to correct these problems in the future.

Hydrological data show that 2017 was colder than 2016. The winter convection in the Black Sea is driven by cooling, and the temperature is used as an indicator of the convection in many previous studies (please, see the comment below) Anomalously cold winter and intense convective mixing in 2017 compare to other years in the 2010-2020 period was already documented in several previous studies (Stanev et al., 2019; Capet et al., 2020). Stanev et al., 2019 showed that the cold winter of 2017 causes intense ventilation of the cold intermediate layer in 2017. A study of dissolved oxygen variability in 2017 was already done by Capet et al., 2020. Capet et al., 2020 show that oxygen content was highest in the winter of 2017 due to strong cooling and convective mixing. In our manuscript, we just confirm this already documented fact (about cold winter and strongest convection in 2017) to give an oceanographic context for the interpretation of the bio-optical properties.

We have underlined this fact in the revised manuscript at lines 206-208: "To conclude, the above analysis is used to argue that the vertical entrainment of nutrients from deep isopycnal layers was more intense in the cold winter of 2017 than in the warm winter

of 2016. This fact is in agreement with recent studies based on the analysis of T, S-diagrams (Stanev et al., 2019) and oxygen variability (Capet et al., 2020) from Argo measurements."

GC5. "If convective mixing is indeed present in winter of 2017, then one should be able to see strong cooling events preceding the mixing events. For this one could perhaps use a reanalysis product or something similar. The heat flux could even be estimated for these cooling events, although it may be enough to correlate temperature anomalies with the mixing events. If there are indeed strong cooling events preceding the mixing, then this would certainly strengthen the argument".

Answer GC5. Convective mixing in the Black Sea is observed every year. It is a subject of investigations in many amount of previous studies in the basin, for example:

Staneva, J.V., Stanev, E.V.: Cold Intermediate Water Formation in the Black Sea. Analysis on Numerical Model Simulations. In: Özsoy E., Mikaelyan A. (eds) Sensitivity to Change: Black Sea, Baltic Sea and North Sea. NATO ASI Series (Series 2: Environment), 27, Springer, Dordrecht, https://doi.org/10.1007/978-94-011-5758-2_29, 1997.

Ivanov, L. I., Backhaus, J. O., Özsoy, E., & Wehde, H. (2001). Convection in the Black Sea during cold winters. Journal of marine systems, 31(1-3), 65-76.

Titov, V. B., 2004. Formation of the upper convective layer and the cold intermediate layer in the Black Sea in relation to the winter severity. Oceanology 44: 327–330.

Oguz, T., Dippner, J. W., & Kaymaz, Z. (2006). Climatic regulation of the Black Sea hydro-meteorological and ecological properties at interannual-to-decadal time scales. Journal of Marine Systems, 60(3), 235-254.

Belokopytov, V. N. (2011). Interannual variations of the renewal of waters of the cold intermediate layer in the Black Sea for the last decades. Physical Oceanography, 20(5), 347-355. Piotukh, V. B., Zatsepin, A. G., Kazmin, A. S., & Yakubenko, V. G. (2011). Impact of the Winter Cooling on the Variability of the Thermohaline Characteristics of

the Active Layer in the Black Sea.Oceanology, 51(2), 221.

Korotaev, G. K., Knysh, V. V., & Kubryakov, A. I. (2014). Study of formation process of cold intermediate layer based on reanalysis of Black Sea hydrophysical fields for 1971-1993. Izvestiya. Atmospheric and Oceanic Physics, 50(1), 35.

In the major part of these studies, it is indicated that thermal conditions play the main role in the processes of ventilation of waters in the winter period. All these studies also confirm that temperature is a reliable indicator of the intensity of winter convection in the Black Sea. In this study, we do not have a goal to investigate in detail convective processes in the basin. These processes were particularly investigated in many cited studies.

Anomalously cold winter and intense convective mixing in 2017 compare to other years in the 2010-2020 period was already documented in several previous studies (Stanev et al., 2019; Capet et al., 2020). In our manuscript, we just confirm this already documented fact (about cold winter and strongest convection in 2017) to give an oceanographic context for the interpretation of the bio-optical properties.

We have added the comment to the text (lines 100-103): "The thermal conditions plays the main role in the processes of ventilation of waters in the winter period. Therefore water temperature is used as a reliable indicator of winter convection in the Black Sea (see e.g., Blatov et al., 1984; Staneva, Stanev, 1997; Ivanov et al., 2001; Belokopytov & Shokurova, 2005; Knysh et al., 2011; Piotuch et al., 2011 and many others)."

GC6. "I recommend that the authors provide a quantitative estimate of the DCM depth, so that its temporal variability be assessed objectively. I can think of various ways this could be achieved, perhaps by obtaining the mean depth of the 90th or 95th percentile of chl-a concentration for each profile. A time series of the DCM depth could then be produced for both floats and the cold/warm years compared quantitatively".

Answer GC6. Thank You for this good advice. We have added the graphs of the position of the lower boundary of DCM to the revised paper (see Fig. 3c). We subjectively defined DCM as a layer with Chl larger than 0.2 mg/m3. This graph complement our results and demonstrate that DCM in 2016 was deeper than in 2017.

GC7. "The level of English in some parts of the manuscript detracts from the value of the science being presented. I provide some suggestions for specific passages below, however, I would strongly suggest that the authors further edit the manuscript to improve clarity and the communication of the findings".

Answer GC7. According to Your comment, we have carefully checked and corrected English grammar in the revised version of the text.

Specific Comments (SC)

SC0. "All figures: The captions lack detail and in many cases are unclear. I suggest carefully reviewing them, adding additional details and rewording to avoid confusion. I give some examples below, but I suggest to revise all captions".

Answer SC0. We have improved all figures' captions in the revised version of the manuscript according to Your comment.

SC1. "Line 27 (and subsequent use): I'm not sure what is meant by "nitroclyne." Please define this."

Answer SC1. We agree and have changed this word on nutricline in the manuscript.

SC2. "Lines 58-59. Is this really true that: "The amount of Chl and related water clarity largely control the depth of the euphotic zone (Shigesada & Okubo, 1981; Morel, 1991). "What about solar angle, time of year? Non-organic particles? Time of year is mentioned earlier in the text, but here it seems like Chl is essentially the only factor. I would reword to "The amount of Chl and related water clarity strongly impact the depth of the euphotic zone . . ."

Answer SC2. Thank You. We agree and have corrected this sentence.

SC3. "Line 47. What is meant by the term "dynamic upwelling"? Please clarify in the text or reword, since this is not standard terminology".

Answer SC3. We have rewritten this phrase as "such as storms, upwellings, or vertical advection in eddies."

SC4. "Line 62 -63. What is the degree of shoaling of the euphotic zone reported in Letelier et al. (2004)? How is phytoplankton impacted and what is specifically meant by "deep layers" (i.e. how deep)?"

Answer SC4. We have added these details to the text (lines 64-66): "On seasonal time scales, Letelier et al. (2004) have shown that the winter bloom of phytoplankton in the tropical Pacific Ocean leads to the additional shoaling of the euphotic zone on about 20 m, inhibiting the development of phytoplankton in the deep layers (below 100 m depth)."

SC5. "Lines 80-82. "Due to the strong haline stratification, the position of chemical layers in the Black Sea is tightly coupled to certain isopycnals and the variations of their concentration in density coordinates are significantly less than in z-coordinates. "Do you mean that vertical variations in the concentration of certain chemicals is significantly less in density coordinates than in z-coordinates? If so, please state this more clearly since the wording is potentially ambiguous. I would also suggest briefly stating why this is important/ significant".

Answer SC5. We agree and have rephrased this sentence in the revised manuscript (lines 105-106): "That is why the variations of their concentration in isopycnic (or $\sigma$) coordinates are significantly less than in vertical z-coordinates (Konovalov et al., 2005; Tuğrul et al., 2014)."

SC6. "Lines 173 - 179: Do you mean here that large-scale circulation is intensified in cold years? If so, a revision of the wording is needed to make this clear. In addition, you would need to describe this phenomenon in more detail (i.e. what is the mechanism?)".

Answer SC6. Thank You. We agree with this comment and have corrected this paragraph (lines 112-116): "Usually, the most intense cyclonic circulation in the basin is observed in the cold years (Blatov et al., 1984; Oguz et al., 2006). Both cyclonic circulation and winter cooling are driven by the same atmospheric patterns – intensification of northeast winds bringing cold continental air from Eurasia (Kubryakov, Stanichny et al., 2019). The rise of cyclonic circulation causes uplift of the pycno-halocline and bring nutricline closer to the surface, but on the opposite decreases MLD (Titov, 2004)."

SC7. "Lines 223 - 229: This passage is currently very unclear. What negative anomalies are the authors referring to? Do they mean the negative values shown in Figure 6e and f? In that case, they should not be referred to as anomalies (which suggest a difference with respect to a long term mean) but as differences (higher or lower chl-a in 2017/2016) or perhaps just "negative values." I would suggest revising these lines, making clear what features the authors refer to and in which figure panels. The authors also suddenly start talking about the geographical location of the 2 floats, without any preamble or reference to Figure 1. I suggest to remind the reader of the location and trajectory of the 2 floats before discussing chl features detected by each".

Answer SC7. We agree and have changed "anomalies" on "values" in the text. We have also rephrased this paragraph (lines 283-288): "There are also noticeable differences in the float measurements, which were possibly caused by the differences in their geographical location. High Chl values measured by the float #7900591 in the central part of the basin were located in a relatively narrow layer with a thickness of 20 m (Fig. 7a, b). Chl distribution measured by the float #6901866 over the continental slope was characterized by a larger thickness of DCM in both years (Fig. 7c, d). The strongest and widest negative differences of Chl were detected by float #6901866 in the whole 35-75 m in July-September, while float #7900591 detected such values throughout the whole season in the narrower layer with a thickness of 20 m (Fig. 7e, f)".

SC8. "Line 244: What is meant by "compensational irradiance"? I suggest to clarify in

the text".

Answer SC8.    The compensation irradiance is the irradiance at which gross planktonic primary production equals to respiration.    Below this irradiance, the demand for respiration exceeds the production by photosynthesis, Chl rapidly decreases.    Phytoplankton became heterotrophic or die off (see e.g.    Regaudieâ ĂŘdeâ ĂŘGioux, A., & Duarte, C. M. (2010)).    Compensation irradiance for planktonic community metabolism in the ocean.    Global biogeochemical cycles, 24(4),    https://agupubs.onlinelibrary.wiley.com/doi/full/10.1029/2009GB003639).    We have rephrased this paragraph (lines 300-303): "The euphotic zone is marked in Fig. 4a, b, Fig. 5a, b as the isolume $E_d=3$ $\mu$mol of photons m-2 s-1 (or 0.08 mmol photons m-2 day-1).    Below this isolume Chl rapidly declines in the Black Sea (Kubryakov et al., 2020). This indicates that this isolume can play a role of the compensation irradiance in the basin (the irradiance at which gross planktonic primary production equals to respiration)."

SC9. "Figure 8: I don't think it's that useful to have the NO3 depicted in both panels of the figure if the profile is exactly the same".

Answer SC9.  We agree and have revised this figure to qualitatively demonstrate the changes in NO3 distribution in a warm and cold year.

Technical Comments (TC)

TC1. "Line 35: "The biomodelling study by Kubryakova et al. (2018)" → I would not use the word "biomodelling," this is definitely not a standard term that is recognised by the community. Biogeochemical or ecosystem model would be more appropriate (or just "modelling")".

Answer TC1. "Biomodelling" was replaced by "modelling".

TC2. "Line 45: "nutrients" should be nutrient."

Answer TC2. Thank You. Corrected.

[Figure]

TC3. "Line 54: change "documented for the Black Sea in ... "to "which has been documented in the Black Sea (references)."

Answer TC3. This phrase is corrected.

TC4. "Throughout the manuscript please change "buoys" to floats. The use of buoys may lead to confusion since BGC-ARGO are floats".

Answer TC4. Throughout the manuscript, the word "buoys" was replaced by "floats."

TC5. "Figure 1: I suggest to only show the isobaths that are labelled (2000, 1600, 1000, 200 m), since as the figure is now there are so many that it becomes meaningless".

Answer TC5. We agree and have corrected the Fig. 1.

Figure 1: Trajectories of the floats #6901866 and #7900591 in 2016 and 2017. The colorbar shows the date of the measurements. Purple crosses – the start of the trajectories, black crosses –end of trajectories. Gray lines show the position of isobaths.

TC6. "Line 125: What is the depth of the reference density used for the MLD calculation?"

Answer TC6. The reference density was taken at 1 m depth. We have added this information to the text (line 157).

TC7. "Figure 4: Which float is the data taken from? If it is an interpolation of both then the method of interpolation must be provided. Add details to the caption".

Answer TC7. This figure shows monthly-averaged values obtained from the average data of two floats. No interpolation was used. We have added this information to the caption.

TC8. "Figure 5: State in the caption how the difference is computed, is it 2016 - 2017 or the other way around? Following this, it would also be helpful to say what positive and negative values mean, e.g. "positive values indicate the chl values are higher in

2017".

Answer TC8. Thank You. We have added this information in the caption for Fig. 6 and 7.

TC9. "Figure 7: It is unclear what is being compared here. Are the red lines 2016 and blue 2017? Or do they represent different floats? Please clarify in the caption, and also add legends to the figures.

Answer TC9. Thank you. We have added this information to the caption: red line – 2016, blue line – 2017.

TC10. "Line 154: conventional should be convectional".

Answer TC10. Thank You. Corrected.

TC11. "Line 213: "Ten-daily diagram. . . "Change to "Fig. 6a-d shows the same features at a higher frequency of 10 days...".

Answer TC11. We have corrected this phrase.

TC12. "Line 233: "Jule-September".

Answer TC12. We agree and have corrected the text.

TC13. "Lines 291 - 292: "Entrained in winter period nutrients and the rise of the ir-radiance causes the following spring growth of phytoplankton. "Reword as: "Winter entrainment of nutrients, followed by increased irradiance in spring, is known to lead to enhanced phytoplankton growth."

Answer TC13. Thank You for this advice. It is corrected.

Please also note the supplement to this comment:
https://bg.copernicus.org/preprints/bg-2020-366/bg-2020-366-AC4-supplement.pdf

[Figure]

Interactive
comment

[Figure]

**Fig. 1.** Fig. R1: Profiles of float #690186 of February 2016 (blue line) and 2017 (red line): (a) – only profiles with MLD higher in 2016 than in 2017; (b) – only profiles with MLD lower in 2016 than in 2017

[Figure]

**Fig. 2.** Fig. R2: Variability of Chl according to Bio-Argo float #6901866: (a) – the Reviewer's image; (b) – Chl visualized using "imagesc" function; (c) – Chl visualized using "contourf" function

[Figure]

**Fig. 3.** Supplementary Fig. S1 (see the caption in the text, answer GC4)

NITRATE on 1m. Trend = 0.234 per year

Fig. 4. Fig. R3: Interannual variability of NO3 at 1 m depth in 2015-2020 according to average data of Bio-Argo floats.

[Figure]

**Fig. 5.** Fig. 3c: Seasonal variability of the lower boundary of DCM (defined as a layer with Chl > 0.2 mg/m3) in 2016 (red line) and 2017 (blue line).

**Fig. 6.** Figure 1 (see the caption in the text, answer TC5)